# Reverse the auditory processing pathway: Coarse-to-fine audio reconstruction from fMRI

## Abstract

Drawing inspiration from the hierarchical processing of the human auditory system, which transforms sound from low-level acoustic features to high-level semantic understanding, we introduce a novel coarse-to-fine audio reconstruction method. Leveraging non-invasive functional Magnetic Resonance Imaging (fMRI) data, our approach mimics the inverse pathway of auditory processing. Initially, we utilize CLAP to decode fMRI data coarsely into a low-dimensional semantic space, followed by a fine-grained decoding into the high-dimensional AudioMAE latent space guided by semantic features. These fine-grained neural features serve as conditions for audio reconstruction through a Latent Diffusion Model (LDM). Validation on three public fMRI datasets—Brain2Sound, Brain2Music, and Brain2Speech—underscores the superiority of our coarse-to-fine decoding method over stand-alone fine-grained approaches, showcasing state-of-the-art performance in metrics like FD, FAD, and KL. Moreover, by employing semantic prompts during decoding, we enhance the quality of reconstructed audio when semantic features are suboptimal. The demonstrated versatility of our model across diverse stimuli highlights its potential as a universal brain-to-audio framework. This research contributes to the comprehension of the human auditory system, pushing boundaries in neural decoding and audio reconstruction methodologies.

## 1 Introduction

Hearing is one of the most important senses for humans, responsible for receiving external auditory stimuli and transmitting the information to the brain for processing and understanding. Researchers aim to explore the auditory perception mechanisms of the human brain from the fields of both neuroscience and computer science (Kell et al., 2018; Millet et al., 2022; Tuckute et al., 2023; Giordano et al., 2023; Caucheteux et al., 2023). A key goal is to decode neural information from the human brain and reconstruct the original stimuli. This can be applied to auditory attention decoding (Van Eyndhoven et al., 2016; O'Sullivan et al., 2017), with practical uses in enhancing hearing aids and assisting communication in noisy environments. Additionally, by generalizing from perception tasks to imagination tasks (Horikawa & Kamitani, 2017; Tang et al., 2023), it serves as a foundation for future research on reconstructing imagined audio.

The common brain-to-audio reconstruction tasks can be categorized into *brain-to-sound* task (Santoro et al., 2017; Park et al., 2023) for reconstructing all natural sounds in the environment, *brain-to-music* task (Bellier et al., 2023; Daly, 2023; Denk et al., 2023) for the music, and *brain-to-speech* task (Pasley et al., 2012; Yang et al., 2015; Hassan et al., 2018; Shigemi et al., 2023; Kim et al., 2023; Chen et al., 2024) for the human voice, based on the different stimulus audios.

Some researchers first attempted to map brain signals to the spectrograms or mel-spectrograms of the stimulus audios using linear regression (Pasley et al., 2012; Yang et al., 2015; Hassan et al., 2018; Bellier et al., 2023). Others introduce non-linear units and use simple networks such as MLP (Yang et al., 2015; Hassan et al., 2018; Bellier et al., 2023), BiLSTM (Shigemi et al., 2023; Daly, 2023), Transformer (Shigemi et al., 2023), etc. This approach can restore the overall temporal and fre-

Figure 1: (a) The hierarchical auditory processing pathway of humans. The stimulus audio is gradually decomposed into time-frequency representation, low-level acoustic features, and high-level semantic characteristics. (b) The pipeline for our coarse-to-fine reconstruction from fMRI. Brain activity is decoded progressively into semantic, acoustic, and spectrogram levels, ultimately resulting in reconstructed audio.

quency information of the spectrogram, but the reconstructed audio lacks semantic and detailed information, especially for non-invasive brain signals.

As research progresses, researchers have found that Deep Neural Network (DNN) features are closer to neural responses in the human brain compared to artificial acoustic representations like spectrograms (Kell et al., 2018; Li et al., 2023; Tuckute et al., 2023; Giordano et al., 2023). Therefore, researchers (Park et al., 2023; Kim et al., 2023; Chen et al., 2024) first decode neural signals into DNN features as an intermediate representation and then use generative models to reconstruct the spectrogram. The intermediate representation is typically chosen from the intermediate layers of DNN (Iashin & Rahtu, 2021; Baevski et al., 2020), serving as fine-grained features that contain both semantic and acoustic information of sound. However, decoding the fine-grained features directly is often challenging due to the high dimensionality, yielding limited outcomes in reconstruction.

There are also works that decode neural signals coarsely into the low-dimensional semantic space. For example, Denk et al. (2023) decodes fMRI data into 128-dimensional MuLan (Huang et al., 2022b) embeddings, which are aligned with simple music descriptions in natural language, and then generates music using MusicLM (Agostinelli et al., 2023). Hence, Denk et al. (2023) primarily focuses on decoding the semantic features within the music, while the acoustic details are largely inferred from the MusicLM's priors, resulting in limited reconstruction similarity. In addition, this model struggles to reconstruct audio beyond music and exhibits poor generalization capabilities.

To enhance the fine-grained decoding, we refocus our attention on neuroscience. As shown in Figure 1(a), research has indicated that in the cochlea and subcortical structures of the human ear, sound is decomposed into frequency-specific temporal patterns similar to spectrograms (Pickles, 1988; Shamma & Micheyl, 2010; Schnupp, 2011; Moore, 2012). Further into the cerebral cortex, the human auditory system has two information processing pathways from low-level to high-level (Rauschecker & Tian, 2000; Kaas & Hackett, 2000; Scott & Johnsrude, 2003; Hickok & Poeppel, 2007; Rauschecker & Scott, 2009). In recent years, an increasing amount of research has found that this cortical processing hierarchy aligns with the functional hierarchy of auditory DNN (Güçlü et al., 2016; Kell et al., 2018; Vaidya et al., 2022; Millet et al., 2022; Li et al., 2023; Tuckute et al., 2023; Giordano et al., 2023). The primary auditory cortex is more sensitive to shallow or intermediate DNN features, which represent low-level acoustic features, while the nonprimary auditory cortex is more sensitive to deep DNN features, which represent high-level semantic features.

Inspired by the acoustic-to-semantic stream, we model each physiological structure of the auditory processing pathway and propose an opposite coarse-to-fine audio reconstruction method, as shown in Figure 1(b). We use non-invasive fMRI as the neural signal. To ensure generalizability across diverse audio stimuli, we choose pretrained models designed for broad applications: CLAP (Wu et al., 2023), a contrastive audio-language model that maps audio to a semantic space; AudioMAE (Huang et al., 2022a), a self-supervised audio model for learning rich acoustic representations; and a Latent Diffusion Model (LDM)(Rombach et al., 2022), which enables high-quality generation. First, we conduct a coarse-to-fine brain decoding process, decoding fMRI data into the low-dimensional CLAP space for coarse-grained semantic features and further into the high-dimensional AudioMAE

latent space for fine-grained acoustic features. Next, the fine-grained features guide mel-spectrogram reconstruction via LDM, with the final waveform restored by a Vocoder(Kong et al., 2020a).

We validate our approach on three publicly available fMRI datasets with different types of audio stimuli: Brain2Sound (Park et al., 2023), Brain2Music (Nakai et al., 2022), and Brain2Speech (LeBel et al., 2023) datasets. Our model achieves state-of-the-art levels in metrics such as FD, FAD, and KL. Experimental results show that our coarse-to-fine framework significantly enhances the decoding of fine-grained audio embeddings and performs well across various datasets, showcasing the potential as a general framework.

Analyzing the coarse-to-fine decoding process, we find that poor semantic decoding can weaken the semantic information in fine-grained embeddings. Given the complexity of sound signals and the low resolution of neural data (Park et al., 2023), brain-to-audio decoding, especially semantic decoding, is highly challenging. To address this, we provide simple semantic prompts (e.g., music genres or speaker genders) during decoding. Experiments show that prompts enhance the semantic quality of reconstructed audio when decoded features are insufficient.

Our contributions are as follows: (1) We propose a coarse-to-fine neural decoding model and reconstruct high-quality waveforms with both semantic and detailed information. We also confirm that coarse-to-fine decoding is superior to solely fine-grained decoding. (2) Our model achieves good results on datasets with three different kinds of stimuli, demonstrating its strong transferability. It can serve as a universal brain-to-audio framework. (3) We attempt to provide semantic prompts and prove that they can enhance the reconstruction quality when semantic decoding is challenging. The code is anonymously released at `https://anonymous.4open.science/r/C2F-LDM`.

## 2 METHOD

Let $y \in \mathbb{R}^L$ represent an audio stimulus and $x \in \mathbb{R}^V$ represent the corresponding fMRI signal, where $L$ is the length of the audio samples and $V$ is the number of voxels in $x$. The brain-to-audio reconstruction process can be formulated as $\mathcal{R} : x \mapsto y$. Our approach is to first decode an intermediate representation $c$ from $x$, and then generate $y$ using a generative model $\mathcal{G}$ conditioned on $c$. To obtain the condition $c$, we follow a coarse-to-fine process. First, we perform a coarse-grained decoding by a Semantic Decoder $\mathcal{D}^{Sem} : x \mapsto s$ to extract the semantic embedding $s$ from fMRI. Then, we use a semantically-guided Acoustic Decoder $\mathcal{D}^{Aco} : (s, x) \mapsto c$ to jointly decode the condition c with both semantics and acoustic details. After decoding, we use an LDM as the generative model $\mathcal{G} : c \mapsto y$ to reconstruct the stimulus audio conditioned on $c$. We will introduce the coarse-grained decoding process of $\mathcal{D}^{Sem}$ in Section 2.1.1, discuss the design of $\mathcal{D}^{Aco}$ and the fine-grained decoding process in Section 2.1.2, and describe the training of $\mathcal{G}$ in Section 2.2.

### 2.1 COARSE-TO-FINE BRAIN DECODING

#### 2.1.1 COARSE-GRAINED SEMANTIC DECODING

We use the CLAP feature as the coarse-grained semantic embedding of audio. CLAP, or contrastive language-audio pretraining (Wu et al., 2023), is a pretrained multi-modal model that aligns representations of audio with natural language descriptions. Pretrained on LAION-Audio-630K dataset (Wu et al., 2023) containing audios of human speech and song, natural sounds, and audio effects music, CLAP features are semantically aligned with various categories of audios, providing rich semantic information.

We model the Semantic Decoder $\mathcal{D}^{Sem} : x \mapsto s$ as a ridge regression model. As shown in Figure 2, we firstly use the final-layer feature of CLAP's Audio Encoder as the ground truth semantic feature of the stimulus audio $y$, denoted as $s_{gt} \in \mathbb{R}^{512}$. Then, we perform the L2-regularized linear regression from $x$ to $s_{gt}$ using PyFastL2LiR[1] toolkit, which provides fast ridge regression and voxel selection functionalities. For each dimension of $s_{gt}$, we only select 500 voxels for regression based on the correlation coefficient. Thus, we obtain a sparse mapping matrix $W \in \mathbb{R}^{V \times 512}$ and a bias $b \in \mathbb{R}^{512}$. The semantic embedding $s$ of fMRI can be inferred by $s = xW + b$ and $s \in \mathbb{R}^{512}$.

---

[1]https://github.com/KamitaniLab/PyFastL2LiR

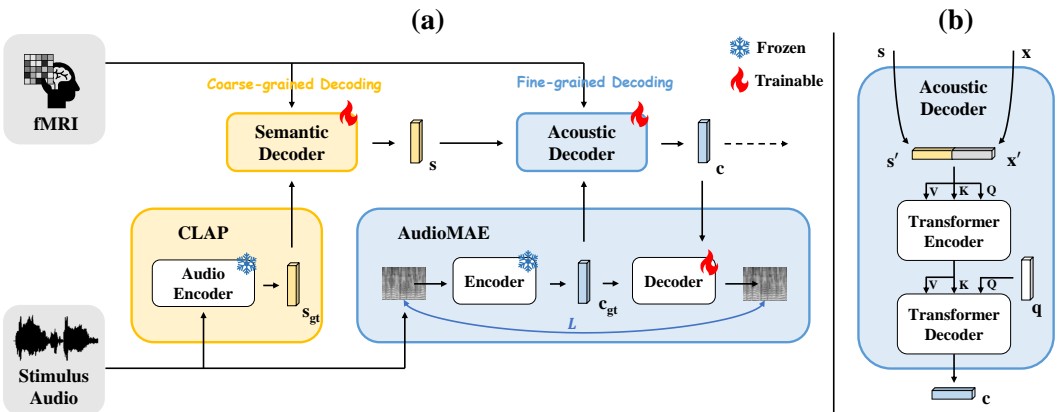

Figure 2: (a) Coarse-to-fine brain decoding. In the **coarse-grained decoding**, fMRI is decoded into the semantic space of CLAP. In the **fine-grained decoding**, fMRI is decoded into the acoustic space of AudioMAE. (b) Detailed structure of Acoustic Decoder.

### 2.1.2 FINE-GRAINED ACOUSTIC DECODING

We use the AudioMAE latent feature as the fine-grained acoustic embedding of audio. AudioMAE, or audio mask autoencoder (Huang et al., 2022a), is a self-supervised pretrained model, which consists of an encoder $\mathcal{E}^A$ and a decoder $\mathcal{D}^A$ and focuses on the reconstruction of the masked patches.

The reason we choose the AudioMAE latent embedding as the acoustic feature instead of other DNN features is threefold: (1) AudioMAE is trained on a generative task, which retains more low-level acoustic details compared to the discriminative models like VGGish-ish (Iashin & Rahtu, 2021) used in Park et al. (2023). (2) Compared to the normal autoencoder used in Chen et al. (2024), AudioMAE performs a masked patch prediction task, which models the whole patches of the spectrogram. The empirical evidence (Liu et al., 2023b) shows that this makes the AudioMAE feature space more inclined to cluster audio of the same category together compared to VAE, indicating that AudioMAE better preserves high-level semantic information. (3) Pretrained on AudioSet (Gemmeke et al., 2017) which consists of natural sounds, human and animal sounds, and music, AudioMAE can work well in the general audio domain. In comparison, the MuLan (Huang et al., 2022b) used in Denk et al. (2023) and Wav2Vec 2.0 (Baevski et al., 2020) used in Kim et al. (2023) can solely be utilized for music or speech. Considering all the points mentioned above, AudioMAE features are highly suitable for fine-grained features in our method, containing rich semantic and acoustic details.

As shown in Figure 2, we first transform the stimulus waveform $y$ into 128 Kaldi (Povey et al., 2011)-compatible Mel-frequency bands with a 25ms Hanning window that shifts every 10 ms following AudioMAE (Huang et al., 2022a), obtaining the mel-spectrogram $m$. Then we divide $m$ into $16 \times 16$ patches $m^p \in \mathbb{R}^{N_{patch} \times 256}$ and encode the patches into $c_{gt} = \mathcal{E}^A(m^p) \in \mathbb{R}^{N_{patch} \times 768}$ with no mask, where $N_{patch}$ represents the number of patches. $c_{gt}$ is then decoded into the reconstructed patches $m^p_{upp} = \mathcal{D}^A(c_{gt})$ and unpatchified into the mel-spectrogram $m_{upp}$. We consider $c_{gt}$ as the ground truth acoustic feature of the stimulus audio $y$ and $m_{upp}$ as an upper bound for the reconstructed mel-spectrogram.

We model the Acoustic Decoder $\mathcal{D}^{Aco} : (s, x) \mapsto c$ as a Transformer-based model, which captures the dependencies between $s$ and $x$, and decodes fMRI into the latent space of AudioMAE through a Seq2Seq generation. First, we project $s$ and $x$ into the 768-dimensional representation space of the Transformer. For $s$, we use a linear layer to project it to a semantic token $s'$. For $x$, we select 768 voxels with the highest responses based on the mapping matrix $W$, forming the fMRI token $x'$. Then we concat the tokens and encode them with a Transformer Encoder $\mathcal{E}^T$, obtaining the neural embedding $n = \mathcal{E}^T([s', x'])$. We create a learnable embedding $q$ as the query to a Transformer Decoder $\mathcal{D}^T$ along with $n$ as key and value, obtaining the decoded acoustic feature $c = \mathcal{D}^T(q, n)$.

**Losses.** We train $\mathcal{D}^{Aco}$ from scratch with three different loss functions that measure the distance between $c$ and $c_{gt}$. The first one is $\mathcal{L}_{cond}$, which directly calculates the L2 distance in the la-

Figure 3: Brain-to-audio reconstruction. The LDM generates mel-spectrograms under the condition of fine-grained acoustic features, followed by the Vocoder to generate reconstructed audios.

tent space. Then, we use the AudioMAE Decoder $\mathcal{D}^A$ to decode $c$ into the reconstructed patches $m_{recon}^p = \mathcal{D}^A(c)$ and unpatchify them into the mel-spectrogram $m_{recon}$. During the decoding process, we calculate the L2 distance between each intermediate layer representation of $\mathcal{D}^A$ and the ground truth, which is the perceptual loss $\mathcal{L}_{perceptual}$. Finally, we calculate the L2 distance between $m_{upp}$ and $m_{recon}$ with the original mel-spectrogram $m$ as the reconstruction loss $\mathcal{L}_{mel}$. The overall loss is given by:

$$\mathcal{L} = \overbrace{\|c - c_{gt}\|_2^2}^{\mathcal{L}_{cond}} + \overbrace{\sum_{i \in layer} \|\mathcal{D}_i^A(c) - \mathcal{D}_i^A(c_{gt})\|_2^2}^{\mathcal{L}_{perceptual}} + \overbrace{\|m_{upp} - m\|_2^2 + \|m_{recon} - m\|_2^2}^{\mathcal{L}_{mel}}. \quad (1)$$

The pretrained AudioMAE is accustomed to handling masked patches, whereas our method leverages all patches to retain essential acoustic information for reconstruction. Therefore, we freeze $\mathcal{E}^A$ and fine-tune the parameters of $\mathcal{D}^A$ to optimize the reconstruction performance.

Furthermore, we follow Liu et al. (2023b) by setting a $P_{gt} = 0.25$ during training, which means that $\mathcal{D}^{Aco}$ has a 0.25 probability of receiving the ground truth semantic feature $s_{gt}$ as input and a 0.75 probability of receiving the decoded semantic feature $s$ from $\mathcal{D}^{Sem}$. This trick helps reduce the impact of decoding noise and improve the stability of the reconstruction by bringing the decoded space closer to the original audio feature space. We will discuss it in Section 3.4.

## 2.2 Brain-to-audio reconstruction

In this section, we use a generative model $\mathcal{G} : c \mapsto y$ to reconstruct the stimulus audio conditioned on $c$. When performing fine-grained decoding, although we use the AudioMAE Decoder to reconstruct the mel-spectrogram, it is not suitable to serve as the generative model for our method. We will discuss this in detail in Section A.4. Instead, we model the process with a Latent Diffusion Model (LDM) (Rombach et al., 2022). LDM is a powerful generative model that can model complex data distributions in the latent space. It has been extensively used in the audio generation task, such as AudioLDM (Liu et al., 2023a), AudioLDM2 (Liu et al., 2023b) and DiffVoice (Liu et al., 2023c).

We follow the formulation in AudioLDM2 (Liu et al., 2023b) to implement the LDM. As shown in Figure 3, we first use a Hanning window with 64 frequency bins, a window size of 1024, and a hop size of 160 to convert the stimulus audio into the mel-spectrogram. Then compress it to a latent representation $z$ using a VAE. The forward diffusion process is a $T$ steps Markov chain that gradually adds Gaussian noise as

$$q(z_t|z_{t-1}) = \mathcal{N}(z_t; \sqrt{1 - \beta_t} z_{t-1}, \beta_t \mathbf{I}) \quad (2)$$

where $\beta_t$ is a variance schedule. Then the distribution of $z_t$ given $z_0$ can be formulated as

$$q(z_t|z_0) = \prod_{s=1}^{t} q(z_s|z_{s-1}) = \mathcal{N}(z_t; \sqrt{\alpha_t} z_0, (1 - \alpha_t)\mathbf{I}) \quad (3)$$

where $\alpha_t = \prod_{s=1}^{t}(1 - \beta_s)$. The distribution of $z_T$ at the final step will be a standard Gaussian distribution (Ho et al., 2020). The LDM learns a reverse denoising process from the prior distribution

$\mathcal{N}(\mathbf{0}, \mathbf{I})$ to the data distribution $z$ conditioned on $c$. The loss function (Ho et al., 2020; Rombach et al., 2022) in our method can be given as

$$\mathcal{L} = \mathbb{E}_{z_t, \epsilon \sim \mathcal{N}(\mathbf{0}, \mathbf{I}), t \sim \{1, ..., T\}}[\|\epsilon_\theta(z_t, t, c) - \epsilon\|_2^2 + \|\epsilon_\theta(z_t, t, c_{gt}) - \epsilon\|_2^2] \tag{4}$$

where $\epsilon_\theta$ is the denoising network, for which we utilize a Transformer-UNet (T-UNet) following AudioLDM2 (Liu et al., 2023b). After the LDM reconstructs the mel-spectrogram, it will be converted to the waveform using a pretrained HiFiGAN (Kong et al., 2020a) vocoder. We initialize the LDM with pretrained weights from AudioLDM2 and fine-tune $\mathcal{D}^{Aco}$ and the T-UNet during training, while keeping other weights frozen.

## 2.3 CONDITIONAL RECONSTRUCTION

In practical applications, brain-to-audio reconstruction is not always unconditional. Firstly, compared to images, audios exhibit strong temporal correlations. If a subject listens to a long segment of stimulus audio but only a portion needs to be reconstructed using brain signals, considering that other audio segments and the target segments may be semantically similar, they can serve as conditions to provide additional semantic information. Secondly, we may know in advance the coarse-grained category (e.g., human speech or animal sound) of the audio to be reconstructed. Given the challenge of semantic decoding from fMRI (see Section 3.3 for details), we can use the coarse-grained category as the semantic prior to guide the reconstruction process. Thus, we attempt to provide semantic prompts in the form of audio or text to our model for conditional reconstruction, to assess whether it can enhance the quality of the reconstructed audio.

The conditional reconstruction process is straightforward. We utilize CLAP's Text Encoder and Audio Encoder to extract the semantic embedding $s_{prompt}$ of the text prompt and audio prompt. Then we replace $s$ with $s_{prompt}$ as input to $\mathcal{D}^{Aco} : (s_{prompt}, x) \mapsto c$, to obtain the fine-grained acoustic embedding $c$. Finally, we use $\mathcal{G} : c \mapsto y$ to reconstruct the stimulus audio conditioned on $c$.

# 3 EXPERIMENTS

## 3.1 EXPERIMENTAL SETTINGS

**Datasets.** We use three publicly available fMRI datasets to validate our method's performance across different kinds of stimuli: Brain2Sound (Park et al., 2023), Brain2Music (Nakai et al., 2022), and Brain2Speech (LeBel et al., 2023) datasets. Brain2Sound Dataset comprises fMRI signals from five subjects listening to 4-second natural sounds, including human speech, animal, musical instrument, and environmental sounds. The dataset consists of 14,400 training samples and 150 test samples. Brain2Music Dataset comprises fMRI signals from five subjects listening to 1.5-second music clips, consisting of 4,800 training samples and 600 test samples. Brain2Speech Dataset comprises fMRI signals from seven subjects listening to 2-second voice segments, consisting of 9,137 training samples and 595 test samples. For detailed information about the datasets and the preprocessing methods, please refer to section A.1.

**Metrics.** We use PCC (Pearson Correlation Coefficient) and PSNR (Peak Signal-to-Noise Ratio) to measure the similarity between the mel-spectrograms of reconstructed audio and stimulus audio, evaluating the low-level fidelity quality. In addition, we use FD, FAD, KL, and CLAP score, which are commonly employed in audio generation tasks, to evaluate the high-level perceptual quality of the reconstructed audio. FD (Fréchet Distance) calculates the distance in features between generated samples and target samples, extracted from an audio classifier PANNs (Kong et al., 2020b). KL (Kullback–Leibler divergence) calculates the KL divergence of classification logits based on PANNs. FAD (Fréchet Audio Distance) is similar to FD, but it uses VGGish (Kilgour et al., 2018). CLAP score calculates the cosine similarity of CLAP (Wu et al., 2023) embeddings. In our experiments, each subject is trained and tested individually, and the metrics are averaged across subjects.

**Comparison Models.** We compare the reconstruction results of three methods: (1) The direct decoding methods, which map fMRI signals to mel-spectrograms, including a linear regression model (Pasley et al., 2012; Yang et al., 2015; Hassan et al., 2018; Bellier et al., 2023) implemented through Ridge in sklearn, a three-layer MLP (Yang et al., 2015; Hassan et al., 2018; Bellier et al.,

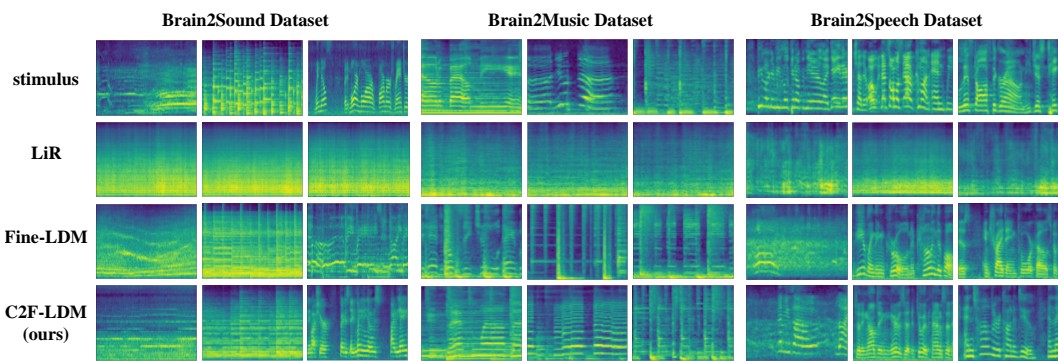

Figure 4: Reconstruction results of S1, sub-001 and UTS01 on the three datasets.

2023) implemented through MLPRegressor in sklearn following Bellier et al. (2023), a Bidirectional LSTM (Shigemi et al., 2023; Daly, 2023) and a Transformer Encoder (Shigemi et al., 2023), both with the same configuration as our Transformer. (2) The fine-grained decoding methods, which map fMRI signals to high-dimensional intermediate features directly (Park et al., 2023; Kim et al., 2023; Chen et al., 2024). We remove the coarse-grained decoding process of our method and decode fMRI into the latent space of AudioMAE using the Acoustic Decoder $\mathcal{D}^{Aco} : x \mapsto c$. Then we used the LDM $\mathcal{G} : c \mapsto y$ to reconstruct the audio. This method is called *Fine-LDM*. In addition, for the Brain2Sound Dataset, we use the code and checkpoints open-sourced by Park et al. (2023) to reproduce their experimental results. (3) The coarse-to-fine decoding methods proposed by us, including *C2F-Decoder*, which utilizes the AudioMAE Decoder as the generative model (see details in Section A.4) and *C2F-LDM* using the LDM (ours). Please refer to section A.2 for specific details on the experimental setup.

## 3.2 RECONSTRUCTION RESULTS

All reconstruction results are presented in Table 1 and 2, which are divided into three sections: direct decoding methods, fine-grained decoding methods, and our coarse-to-fine decoding methods. We select one representative from each section, *Linear Regression* (*LiR*), *Fine-LDM* and *C2F-LDM* to display the reconstructed mel-spectrograms[2] in Figure 4.

We find that direct decoding methods, which are optimized based on mean squared error, can achieve higher PCC and PSNR. However, as shown in Figure 4, the reconstruction results are often overly smooth and lack high-frequency details,

Table 1: Reconstruction results on the Brain2Sound Dataset. **Bold** indicates the best, and underlined indicates that our method outperforms the fine-grained decoding methods.

| Brain2Sound Dataset (Park et al., 2023) | | | | | | |
|---|---|---|---|---|---|---|
| Model | PCC↑ | PSNR↑ | FD↓ | FAD↓ | KL↓ | CLAP↑ |
| LiR | **0.607** | 17.506 | 105.113 | 40.877 | 4.027 | 0.175 |
| MLP | 0.566 | 17.310 | 98.358 | 38.045 | 4.020 | 0.164 |
| BiLSTM | 0.580 | 17.381 | 112.031 | 39.895 | 3.948 | 0.180 |
| Transformer | 0.581 | **17.676** | 104.118 | 39.484 | 3.764 | 0.177 |
| Park et al. | 0.394 | 15.406 | 88.456 | 12.694 | **2.251** | 0.268 |
| Fine-LDM | 0.376 | 14.624 | 49.827 | 10.803 | 2.895 | 0.265 |
| C2F-Decoder | 0.595 | 17.385 | 95.565 | 35.775 | 3.748 | 0.179 |
| C2F-LDM (**ours**) | 0.418 | 15.103 | **44.003** | **9.324** | 2.697 | **0.275** |

leading to poor perceptual quality. In contrast, the fine-grained decoding methods exhibit a significant improvement in FD, FAD, KL, and CLAP score. The reconstructed spectrograms contain more time-frequency details, leading to more realistic reconstructed audios and a preliminary reconstruction of semantics. On the other hand, the fine-grained decoding methods fall short in terms of PCC and PSNR. In signal generation and reconstruction, a theoretical trade-off exists between perceptual quality and distortion metrics (such as PSNR) (Blau & Michaeli, 2018). Improving perceptual quality often leads to lower PSNR values, and this trade-off is evident in our reconstruction tasks. The use of generative models aims to achieve reasonable reconstruction accuracy while preserving high perceptual quality. Experimental results show that a PCC of approximately 0.4 is acceptable, considering the balance between perceptual quality and signal fidelity.

---

[2]Reconstructed spectrograms and audios for all subjects can be found in the supplementary material.

Table 2: Reconstruction results on the Brain2Music and Brain2Speech datasets. **Bold** indicates the best, and underlined indicates that our method outperforms the fine-grained decoding methods.

| Model | Brain2Music Dataset (Nakai et al., 2022) | | | | | | Brain2Speech Dataset (LeBel et al., 2023) | | | | | |
|---|---|---|---|---|---|---|---|---|---|---|---|---|
| | PCC↑ | PSNR↑ | FD↓ | FAD↓ | KL↓ | CLAP↑ | PCC↑ | PSNR↑ | FD↓ | FAD↓ | KL↓ | CLAP↑ |
| LiR | 0.637 | 19.353 | 47.710 | 18.247 | 0.997 | 0.223 | 0.511 | 17.500 | 68.146 | 24.988 | 3.483 | 0.112 |
| MLP | 0.591 | 18.886 | 48.980 | 19.895 | 0.732 | 0.200 | 0.409 | 16.389 | 75.174 | 27.983 | 4.153 | 0.094 |
| BiLSTM | 0.628 | 19.078 | 57.030 | 22.673 | 1.008 | 0.209 | **0.526** | 17.688 | 92.172 | 33.442 | 4.187 | 0.074 |
| Transformer | **0.646** | 19.379 | 60.969 | 22.195 | 1.079 | 0.198 | 0.526 | **17.690** | 74.048 | 27.526 | 3.817 | 0.041 |
| Fine-LDM | 0.419 | 15.526 | 6.412 | **1.273** | 0.548 | 0.512 | 0.357 | 14.385 | 12.706 | 4.820 | 0.885 | 0.420 |
| C2F-Decoder | 0.643 | **19.478** | 63.039 | 26.053 | 1.191 | 0.195 | 0.518 | 17.495 | 96.032 | 26.917 | 4.278 | 0.077 |
| C2F-LDM (**ours**) | 0.454 | 15.883 | **6.102** | 1.504 | **0.520** | **0.530** | 0.393 | 15.260 | **9.726** | **4.623** | **0.616** | **0.471** |

In comparison to the fine-grained decoding methods, our coarse-to-fine approach excels in both low-level and high-level metrics. The spectrogram details are closer to the stimulus audio. Our method achieves state-of-the-art performance in FD, FAD, KL, and CLAP score, while also enhancing PCC and PSNR, although it falls short of direct decoding methods. It demonstrates that coarse-to-fine decoding can effectively enhance the quality of reconstruction. Compared to Park et al. (2023), our method significantly improves on PCC, FD, FAD, and CLAP score. The comparison of reconstructed samples can be found in Section A.3. The comparison with *C2F-Decoder* can be found in Section A.4.

We further analyze the impact of our coarse-to-fine method on decoding the fine-grained acoustic features. We compute the PCC between the ground truth and decoded acoustic features for 17 subjects across the three datasets. Then we compare the experimental results between the coarse-to-fine decoding and the directly fine-grained decoding, with a baseline established by directly mapping fMRI to the acoustic features using L2-regularized Linear Regression. As shown in Section A.5, the consistent decoding performance of the PCC across different methods reflects the varying signal quality of the subjects. Across almost all participants, our coarse-to-fine method consistently outperforms the fine-grained method. It suggests that coarse-to-fine decoding can effectively enhance the fine-grained acoustic features widely across the participants.

### 3.3 SEMANTIC ANALYSIS OF ACOUSTIC FEATURES

An intuitive idea is that the introduction of coarse-grained semantic decoding enhances the semantic information in the acoustic features, thereby improving the fine-grained decoding. Is this really the case? We will discuss this issue in this section.

It is generally believed that a representation space with strong semantic information typically clusters samples of the same category while separating those of different categories. Therefore, we conduct a classification experiment using two datasets with clear category labels, the Brain2Music Dataset with the labels of music genres (10 classes) and the Brain2Speech Dataset with the labels of speakers' genders (2 classes). Gender is chosen as the semantic label for human speech primarily because it is the most intuitive and easily annotated attribute within speech semantics. Additionally, we reference the LAION-Audio-630K dataset (Wu et al., 2023) used by CLAP, where gender information serves as a key element in the text captions for human speech. We perform 5-fold cross-validation on the test set, using SVM to classify acoustic features obtained through coarse-to-fine decoding or directly fine-grained decoding. The average classification accuracy measures the semantic information in the acoustic features, with identical experimental conditions ensuring an unbiased comparison.

As shown in Figure 5, the chance levels are $0.1$ and $0.5$ for the two datasets, while the upper bounds are the classification accuracy on the ground truth acoustic features, which are $0.33$ and $0.95$. After introducing the coarse-grained decoding, the classification accuracy for each subject in the Brain2Music Dataset either increases or remains essentially unchanged. However, in the Brain2Speech Dataset, the classification accuracy for some subjects decreases. It suggests that the semantic information in the acoustic features is diminishing, while the coarse-grained decoding primarily enhances the low-level acoustic information.

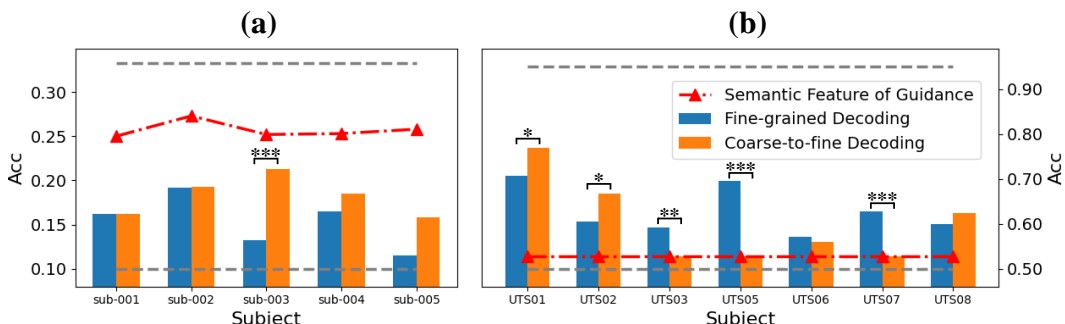

Figure 5: Semantic decoding accuracy in the (a) Brain2Music and (b) Brain2Speech Dataset. The gray dashed lines represent the upper bound of accuracy and the chance level. Significance test is performed (paired t-test, $p < 0.001$(***), $p < 0.01$(**), $p < 0.05$(*)).

To explain this phenomenon, we further analyze the coarse-grained semantic features of guidance. The same SVM classification task is performed. As shown in Figure 5, the classification accuracy of the semantic features in the Brain2Music Dataset is relatively high, whereas the classification accuracy in the Brain2Speech Dataset is poor, approaching the chance level. It indicates that there is little semantic content in the semantic features, making it nearly impossible to differentiate speakers' genders. We visualize the distribution of the semantic features using t-SNE for the two datasets. As shown in Section A.6, some categories in the Brain2Music Dataset cluster well, while samples from the two categories in the Brain2Speech Dataset are mixed and almost indistinguishable, which is consistent with our experimental results. This could be attributed to participants being more focused on the content of the stories during fMRI signal collection, potentially disregarding the speaking style of the speaker. The lack of semantic richness in the guided semantic features leads to a decrease in the semantic content of the acoustic features.

In summary, although introducing the coarse-grained semantic decoding can enhance the decoding of the fine-grained acoustic features, the semantic content of the acoustic features may not be enhanced if the semantic features of guidance are poor. In such cases, the coarse-grained decoding mainly enhances the low-level acoustic information.

## 3.4 CONDITIONAL RECONSTRUCTION RESULTS

In the previous section, we note that poor semantic decoding can degrade the semantic content of acoustic features during guided decoding. Therefore, in the conditional reconstruction task mentioned in Section 2.3, it is preferable to use prompt-based semantic features as coarse-grained features instead of decoded ones. We define two types of prompts: (1) Text prompts, which specify the stimulus audio category. For the Brain2Music Dataset, these include 10 music genres (e.g., *pop music*, *rock music*), while for the Brain2Speech Dataset, they indicate the speaker's gender (*man speaking* or *woman speaking*). (2) Audio prompts, where the last 10 stimulus audio clips from two test set stories in the Brain2Speech Dataset are used as prompts, with results averaged.

To evaluate conditional brain-to-audio reconstruction, we test the hyperparameter $P_{gt}$ mentioned in Section 2.1.2 under three conditions: $P_{gt} = 0.0$, 0.25 and 0.5. The results are shown in Table 3. We observe that incorporating ground truth semantic features during training improves brain-to-audio reconstruction, but higher $P_{gt}$ values do not always yield better results. For instance, in the Brain2Music Dataset, $P_{gt} = 0.5$ performs worse than $P_{gt} = 0.25$. Since fMRI data is used exclusively during testing, excessively high $P_{gt}$ creates a mismatch between training and testing, reducing reconstruction accuracy. Balancing decoding from fMRI and ground truth, we select $P_{gt} = 0.25$ as the optimal value based on metrics from both datasets.

For the Brain2Music Dataset, the high-level metrics decrease after incorporating the text prompts. It suggests that the text prompts of music genres we provide may not effectively represent the semantic content of the stimulus audio. After adding the text prompts, samples of the same music genre have identical coarse-grained features. Considering the relatively strong semantic decoding in the Brain2Music Dataset as mentioned in Section 3.3, the loss of the individual specificity across samples leads to a decrease in the reconstruction quality, particularly in semantics. In contrast, the

Table 3: Conditional reconstruction results with text prompts and audio prompts.

| $P_{gt}$ | Prompt | Brain2Music Dataset (Nakai et al., 2022) | | | | | | Brain2Speech Dataset (LeBel et al., 2023) | | | | | |
|---|---|---|---|---|---|---|---|---|---|---|---|---|---|
| | | PCC↑ | PSNR↑ | FD↓ | FAD↓ | KL↓ | CLAP↑ | PCC↑ | PSNR↑ | FD↓ | FAD↓ | KL↓ | CLAP↑ |
| 0.0 | no prompt | 0.442 | 15.827 | 6.121 | 1.606 | 0.520 | 0.527 | 0.379 | 14.898 | 11.636 | 4.866 | 0.758 | 0.438 |
| | text prompt | 0.421 | 15.600 | 8.283 | 2.169 | 0.641 | 0.460 | 0.331 | 14.531 | 10.772 | 5.265 | 0.513 | 0.444 |
| | audio prompt | - | - | - | - | - | - | 0.315 | 14.543 | 7.160 | 4.148 | 0.430 | 0.481 |
| 0.25 | no prompt | **0.454** | 15.883 | **6.102** | 1.504 | **0.520** | **0.530** | **0.393** | 15.260 | 9.726 | 4.623 | 0.616 | 0.471 |
| | text prompt | 0.405 | **16.059** | 7.358 | 2.219 | 0.584 | 0.470 | 0.340 | 14.680 | 9.265 | 4.712 | 0.449 | 0.476 |
| | audio prompt | - | - | - | - | - | - | 0.300 | 14.434 | 7.722 | 4.383 | 0.416 | 0.491 |
| 0.5 | no prompt | 0.422 | 15.466 | 6.587 | **1.341** | 0.536 | 0.513 | 0.374 | **15.299** | 7.957 | 4.013 | 0.493 | **0.502** |
| | text prompt | 0.393 | 15.827 | 10.304 | 2.624 | 0.559 | 0.465 | 0.350 | 14.912 | **6.698** | 5.102 | **0.348** | 0.487 |
| | audio prompt | - | - | - | - | - | - | 0.303 | 14.401 | 7.129 | **3.930** | 0.398 | 0.486 |

semantic decoding performs poorly in the Brain2Speech Dataset, so the introduction of both text prompts and audio prompts can significantly enhance the semantics of the reconstructed audio. In summary, when coarse-grained semantic features are suboptimal, conditional reconstruction with prompts can effectively enhance the quality of the reconstructed audio.

## 4 LIMITATIONS AND FUTURE WORK

**Temporal resolution.** Given the advantages of high spatial resolution and high signal-to-noise ratio in non-invasive neural signals, fMRI has been commonly employed in the field of neural encoding and decoding. Research (Santoro et al., 2017) has confirmed that the reconstruction from the BOLD response (TR=2.6s) can exhibit a temporal specificity of about 200 ms, which is adequate for capturing certain semantics and details of the audio. Since then, numerous works (Santoro et al., 2017; Park et al., 2023; Denk et al., 2023) and datasets (Nastase et al., 2021; Li et al., 2021; Nakai et al., 2022; Park et al., 2023; LeBel et al., 2023) have emerged to support research on fMRI-to-audio tasks. However, the limited temporal resolution of fMRI consistently hampers the temporal decoding of audio. The aim of this article is to enhance the reconstruction performance further from a neuroscience perspective. To make a breakthrough in temporal decoding, it is imperative to leverage other neural signals with high temporal resolution, such as EEG and MEG.

**Model and voxel selection.** The main purpose of this article is to illustrate the superiority of hierarchical decoding over direct decoding. Therefore, we build a generic brain-to-audio framework, selecting the most suitable models, CLAP and AudioMAE, without comparing them to other representation models. In the future, we will switch to different models within our framework to attempt further improvement of reconstruction results. Furthermore, we utilize all the voxels of the auditory cortex (AC) in our work. However, there are gradients in the voxels of different brain regions within the AC (Kell et al., 2018; Tuckute et al., 2023; Giordano et al., 2023). In the future, we plan to consider the gradients of voxels to further enhance the hierarchy of information processing.

## 5 CONCLUSION

In this paper, we propose a novel coarse-to-fine audio reconstruction method inspired by the hierarchical processing of the human auditory system. Our method begins by decoding fMRI into the CLAP space to extract coarse-grained semantic features. Subsequently, leveraging these semantic features, we decode fMRI into the AudioMAE latent space to capture fine-grained acoustic features. Next, we use the acoustic features as conditions to reconstruct the stimulus audio using a Latent Diffusion Model. By validating on three diverse fMRI datasets, our method has shown superior performance in brain-to-audio reconstruction compared to previous fine-grained methods. We have illustrated its state-of-the-art capabilities, achieving remarkable results in FD, FAD, KL, and CLAP score. The integration of semantic prompts during decoding further enhances the semantics of reconstructed audio, particularly when dealing with suboptimal semantic decoding. We will discuss the broader impacts of our paper in Section A.7.

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

# A APPENDIX

## A.1 DATASETS

Table 4: ROIs and voxels of the datasets.

| Dataset | ROIs | Subjects | Voxels | Training samples | Test samples |
|---|---|---|---|---|---|
| Brain2Sound (Park et al., 2023) | AC (A1, LBelt, A4, A5, etc.) | S1 | 6,662 | 13,872 | |
| | | S2 | 6,624 | 13,944 | |
| | | S3 | 6,713 | 13,944 | 150 |
| | | S4 | 6,157 | 13,944 | |
| | | S5 | 7,143 | 13,944 | |
| Brain2Music (Nakai et al., 2022) | N/A | sub-001 | 60,784 | | |
| | | sub-002 | 53,927 | | |
| | | sub-003 | 64,700 | 4,800 | 600 |
| | | sub-004 | 61,899 | | |
| | | sub-005 | 53,421 | | |
| Brain2Speech (LeBel et al., 2023) | AC, FFA, OFA, PPA, etc. | UTS01 | 836 | | |
| | | UTS02 | 2,093 | | |
| | | UTS03 | 1,303 | | |
| | | UTS05 | 920 | 9,137 | 595 |
| | | UTS06 | 980 | | |
| | | UTS07 | 1,584 | | |
| | | UTS08 | 1,109 | | |

**Brain2Sound Dataset** As proposed by Park et al. (2023), this dataset[3] records the fMRI signals of five subjects (one female) while they are listening to natural sounds, including human speech, animal, musical instrument, and environmental sounds. fMRI data are acquired using a 3.0-Tesla Siemens MAGNETOM Verio scanner at the Kyoto University Institute for the Future of Human Society. Functional images that cover the entire brain are obtained with TR = 2,000 ms, TE = 44.8 ms, flip angle = 70 deg, FOV = 192 × 192 mm, voxel size=2 × 2 × 2 mm, number of slices = 76 and multi-band factor = 4. We utilize fMRI data preprocessed by Park et al. (2023), primarily involving motion correction, slice time correction, co-registration, BOLD time-series resampling, etc.

The stimuli consist of 1,250 8-s natural sound segments, with 1,200 for the training set and 50 for the test set, selected from the *VGGSound dataset* (Chen et al., 2020). All the sounds are extracted from the videos uploaded to YouTube. To increase the sample number, we preprocess the audio segments in the same way as Park et al. (2023): 4-s sliding windows are utilized with a 2-s stride to extract 3 4-s segments. All audio clips are resampled to 16kHz. During the collection of fMRI signals, each stimulus is repeated four times, resulting in 14,400 samples[4] for the training set (1,200 stimuli × 4 repetitions × 3 samples = 14,400 samples). For the test set, we average the multiple fMRI samples, resulting in 150 samples (50 stimuli × 3 samples = 150 samples).

**Brain2Music Dataset** Following Denk et al. (2023), we use the *music genre neuroimaging dataset*[5] from Nakai et al. (2022), which records the fMRI signals of five subjects (two female) while they are listening to music clips. fMRI data are acquired using a 3.0T MRI scanner (TIM Trio; Siemens, Erlangen, Germany) at the Center for Information and Neural Networks (CiNet), National Institute of Information and Communications Technology (NICT), Osaka, Japan. Functional scanning is performed with TR = 1,500 ms, TE = 30 ms, flip angle = 62 deg, FOV = 192 × 192 mm, voxel size = 2 × 2 × 2 mm and multi-band factor = 4. We utilize fMRI data preprocessed by Denk et al. (2023), encompassing essential steps such as motion correction, template alignment, low-frequency drift removal, response normalization, etc.

---

[3]https://github.com/KamitaniLab/SoundReconstruction

[4]When downloading, we discovered that some audios in the training set were no longer available on YouTube, hence, the amount of training samples is slightly less than 14,400. See Table 4 for details.

[5]https://openneuro.org/datasets/ds003720

The dataset contains music stimuli from 10 genres (blues, classical, country, disco, hip-hop, jazz, metal, pop, reggae, and rock) which are sampled from the *GTZAN dataset* (Tzanetakis & Cook, 2002). A total of 54 15-s music pieces are selected from each genre, with 48 for the training set and 6 for the test set. All music pieces are resampled to 16kHz and segmented into 10 clips of 1.5 seconds each to match the TR of functional scanning. As a result, the dataset consists of 4,800 samples (48 stimuli × 10 genres × 10 samples = 4,800 samples) for training and 600 (6 stimuli × 10 genres × 10 samples = 600 samples) for testing.

**Brain2Speech Dataset** We use the dataset[6] proposed by LeBel et al. (2023). The dataset contains fMRI responses recorded while 7 participants[7] (three female) are listening to 27 complete, natural, narrative stories. fMRI data are collected on a 3T Siemens Skyra scanner at the UT Austin Biomedical Imaging Center. Functional scans are collected with TR = 2.00 s, TE = 30.8 ms, flip angle = 71 deg, multi-band factor = 2, voxel size = 2.6 × 2.6 × 2.6 mm and FOV = 220 mm. We utilize fMRI data preprocessed by LeBel et al. (2023), primarily incorporating motion correction, template creation and alignment, low-frequency drift removal, response normalization, etc.

The stimulus set consists of 27 10–15 minute stories from *The Moth* podcast. We select two stories (*Hang time* by a male speaker and *Where there's Smoke* by a female speaker) as the test set, and the remaining 25 stories are used as the training set. All stories are resampled to 16kHz and segmented into 2-s clips to match the TR of functional scanning. To account for the hemodynamic response, we form a sample pair by combining the fMRI signal of each TR with the stimulus audio clip from 4 seconds ago. We use the last 10 stimulus audio clips from two stories in the test set as audio prompts. These prompts are not used for testing, ensuring that the participants could not have possibly heard the audio prompts in the preceding trials. As a result, the dataset consists of 9,137 samples for training and 595 samples for testing per subject.

A.2 EXPERIMENTAL SETUP

In the stage of coarse-grained decoding, for the Brain2Sound and Brain2Speech datasets, we only utilize voxels from the auditory cortex (AC) area, whereas for the Brain2Music Dataset, we use voxels from the entire brain. The specific brain regions and voxels can be found in Table 4.

In the stage of fine-grained decoding, we utilize a 4-layer Transformer Encoder and Decoder in $\mathcal{D}^{Aco}$ and use the default configuration of AudioMAE (Huang et al., 2022a), initialized with the pretrained weights.[8] The AudioMAE Encoder is a vanilla 12-layer ViT-B, while the AudioMAE Decoder is a 16-layer Transformer with shifted local attention. Since AudioMAE requires 10-second audios ($128 \times 1024$ mel-spectrograms) as inputs, we duplicate the stimulus waveforms to 10 seconds. After encoding with the AudioMAE Encoder $\mathcal{E}^A$, we select the embeddings of the first $N_{patch}$ patches as $c_{gt}$, corresponding to the length of the stimulus audio. We set $N_{patch} = 208$ for the Brain2Sound Dataset, $N_{patch} = 80$ for the Brain2Music Dataset, and $N_{patch} = 112$ for the Brain2Speech Dataset.

In the stage of brain-to-audio reconstruction, we follow the formulation in AudioLDM2 (Liu et al., 2023b) to implement the LDM $\mathcal{G}$ and utilize two checkpoints[9] as the initialization weights: *audioldm2-full* for the Brain2Sound and Brain2Music datasets, and *audioldm2-speech-gigaspeech* for the Brain2Speech dataset.

We use the AdamW (Loshchilov & Hutter, 2017) optimizer to train $\mathcal{D}^{Aco}$ and $\mathcal{D}^A$ with a learning rate of 1e-6, and train $\mathcal{G}$ with a learning rate of 1e-4. We train on the Brain2Sound, Brain2Music, and Brain2Speech datasets with a batch size of 8 for 30, 40, and 30 epoches. All training is completed on a single NVIDIA A100 80GB GPU.

A.3 COMPARISON WITH PARK ET AL. (2023)

We reproduce the experimental results of Park et al. (2023) using the features from the $conv5\_3$ layer of VGGish-ish (Iashin & Rahtu, 2021) and voxels from the entire AC region. The results and the comparison with our method are shown in Figure 6.

---

[6]https://openneuro.org/datasets/ds003020/versions/1.1.1

[7]Subject UTS04 lacks a story, hence it will not be utilized.

[8]https://github.com/facebookresearch/AudioMAE

[9]https://github.com/haoheliu/AudioLDM2

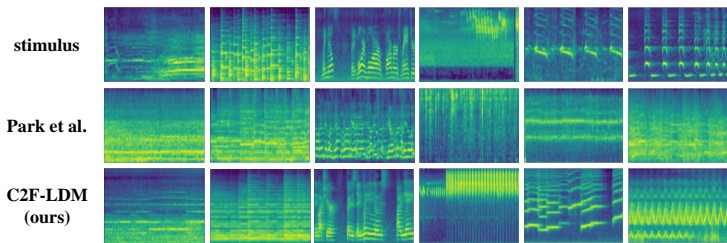

Figure 6: Comparison with the reconstruction results of Park et al. (2023).

## A.4 COMPARISON WITH *C2F-Decoder*

When performing fine-grained decoding, although we use the AudioMAE Decoder to reconstruct the mel-spectrogram, it is not suitable to serve as the generative model for our method. There are two main reasons for this: (1) The mel bins and window parameters of the mel-spectrograms in AudioMAE do not align with those of commonly used pretrained Vocoders. This mismatch prevents the generated mel-spectrograms from being directly converted into audio. Moreover, the cost of training a compatible Vocoder from scratch is prohibitively high. (2) The primary task of the AudioMAE Decoder is to predict masked patches, with a focus on low-level details of the spectrogram. This limitation leads to insufficient reconstruction quality in terms of semantic content. In contrast, the mel-spectrograms generated by LDM can be directly restored to audio using the pretrained HiFiGAN (Kong et al., 2020a) vocoder, and the generated audio has richer semantic and acoustic details.

To investigate the reconstruction performance of *C2F-Decoder*, we need to transform the mel-spectrograms generated by the AudioMAE Decoder, denoted as $m^A$, into mel-spectrograms that the Vocoder can accept, denoted as $m^V$. We assume that $m^A$ and $m^V$ have a linear relationship, so we use an L2-regularized linear regression model trained on $m^A$ and $m^V$ of the stimulus audio in the training set. The results in the test set are as follows: PCC = 0.938 in the Brain2Sound Dataset, PCC = 0.967 in the Brain2Music and Brain2Speech datasets. Based on the results, we believe that this transformation is almost lossless. As shown in Figure 7, Table 1 and Table 2, *C2F-Decoder* is similar to the direct decoding methods in that they both focus on modeling the overall spectrograms but lack details and semantic information compared to *C2F-LDM*. It demonstrates the superiority of LDM over employing the AudioMAE Decoder directly.

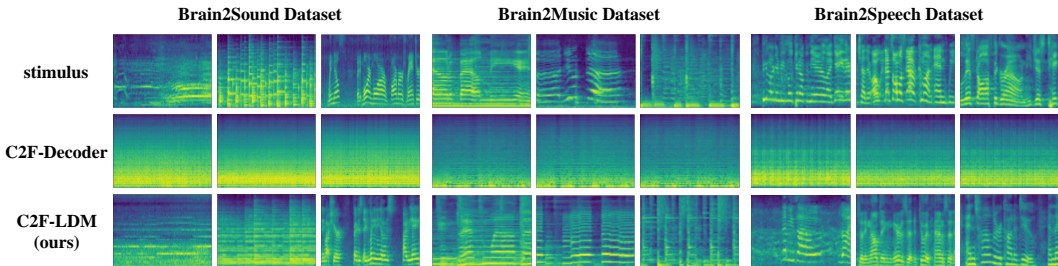

Figure 7: Comparison of the reconstruction results between *C2F-Decoder* and *C2F-LDM*.

## A.5 COMPARISON OF ACOUSTIC DECODING METHODS

We compare the decoding accuracy of acoustic features among three methods: linear regression, fine-grained decoding, and coarse-to-fine decoding. The PCC results for 17 subjects across the three datasets are displayed in Figure 8. Our coarse-to-fine method achieves the highest PCC on two of the datasets and consistently outperforms the fine-grained method across almost all subjects. It indicates that coarse-to-fine decoding can effectively enhance the fine-grained acoustic features widely across the participants.

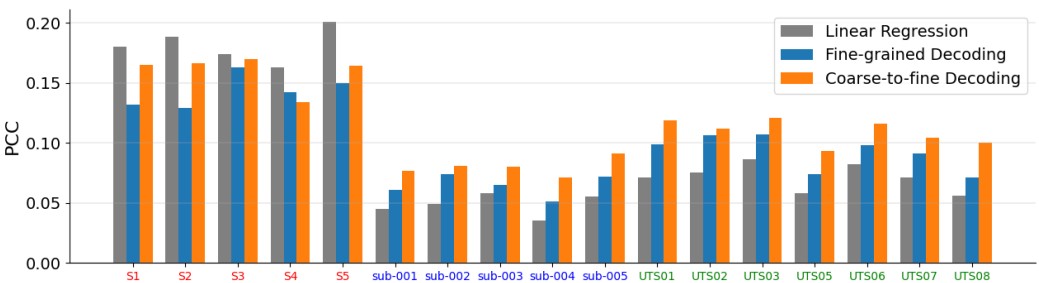

Figure 8: PCC between the ground truth and decoded acoustic features for 17 subjects in the Brain2Sound, Brain2Music and Brain2Speech datasets. Our coarse-to-fine method consistently outperforms the directly fine-grained method.

## A.6 VISUALIZATION OF SEMANTIC SPACE

We perform t-SNE visualization on the space of ground truth and decoded semantic features on the Brain2Music and Brain2Speech datasets. We choose the decoding spaces of sub-003 and UTS05, which show significant effects after incorporating the semantic features based on Figure 5. For the Brain2Music Dataset, some genres like *classical* and *jazz* exhibit good decoding performance, while others like *rock* show poor decoding performance, as shown in Figure 9. For the Brain2Speech Dataset, the semantic decoding performance is poor, and it cannot differentiate between male and female speakers, as shown in Figure 10.

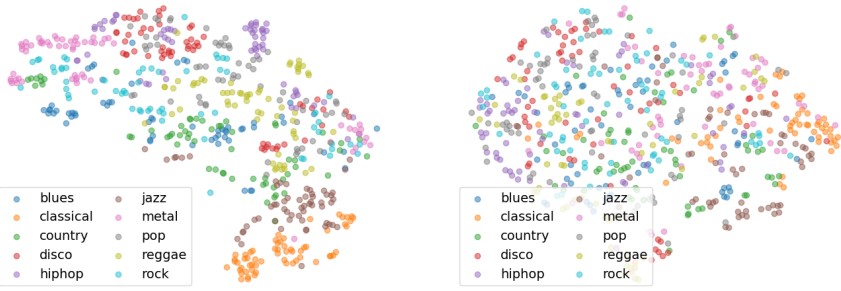

Figure 9: Visualization on the space of ground truth (left) and decoded (right) semantic features on the Brain2Music Dataset.

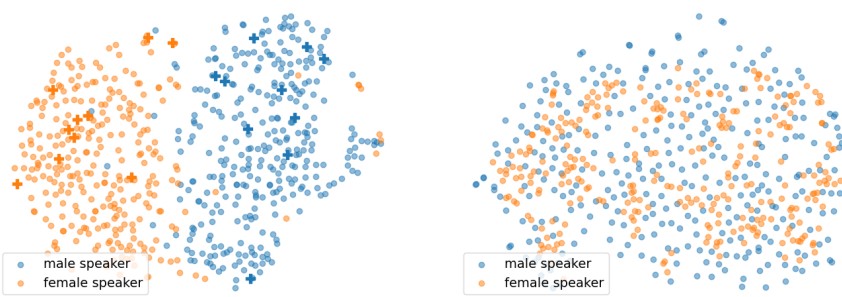

Figure 10: Visualization on the space of ground truth (left) and decoded (right) semantic features on the Brain2Speech Dataset. The cross markers represent samples used as audio prompts, providing coarse-grained semantic features in the conditional reconstruction.

## A.7 BROADER IMPACTS

This research contributes to enhancing our understanding of brain function and cognitive processes, playing a crucial role in further exploring the mechanisms of the human auditory system and promoting the development of related technologies. In contrast to audio tasks like audio generation, speech synthesis, and sound editing, our neural decoding task focuses on reconstructing the audio heard by the subjects. The primary application scenario is a brain-computer interface system designed to aid individuals with voice disorders. This task does not pose potential ethical challenges such as forging or tampering with someone's voice. In addition, the experiments are conducted in a controlled laboratory setting with the participants' consent and cooperation. It is not applicable in real-world scenarios, thereby posing a minimal risk of privacy leakage.

