# OpenReview forum: "Reverse the auditory processing pathway: Coarse-to-fine audio reconstruction from fMRI"
_ICLR.cc/2025/Conference — Submitted to ICLR 2025_

### Official Review · Reviewer_xXUv · 2024-10-26

**Soundness:** 3
**Presentation:** 4
**Contribution:** 3
**Rating:** 8
**Confidence:** 2

**Summary:**

This paper proposes a coarse-to-fine framework for audio reconstruction from fMRI brain recordings that outperforms the leading solely fine-grained approaches. This approach draws inspiration from the hierarchical processing found in the human auditory system. It first projects the audio and fMRI into a coarse-grained semantic embedding in the CLAP (Wu et al, 2023) space before paring that embedding again with the fMRI signal for fine-grained decoding, generating features in the space of AudioMAE (Huang et al, 2022). Lastly, an LDM is used to reconstruct the mel-spectrogram of the stimulus audio with the fine-grained embedding, before it is converted to the waveform using a pretrained HiFiGAN (Kong et al, 2020) vocoder.

The authors evaluate the performance of this framework over three different datasets encompassing three different classes of audio / reconstruction task (sound, music, and speech). The quality of the reconstructed audio is assessed by FD, FAD, KL, and CLAP score and the mel-spectrograms are evaluated using PCC and PSNR. Performance is benchmarked against direct decoding approaches and fine-grained only approaches. The authors also offer an investigation into the quality of the semantic information captured through these approaches.

**Strengths:**

Overall, this paper represents a novel improvement for brain-to-audio decoding and its acknowledgement would benefit the advancement of the field. The approach introduced is novel, performance state-of-the-art, and the presentation clean.

**Weaknesses:**

There are a few places where choices are made without explanation and parts of the discussion on the semantic analysis are unclear. For example, in the discussion of the semantic analysis it is stated that there is less semantic richness in the brain-to-speech case because listeners might be more focused on content. However, this effect does not necessarily seem to hold for the fine-grained only decoding (which is not addressed). It seems more likely that coarse-grained decoding may just be ill-suited to capturing semantic quality from speech as the signal clearly exists given the performance of the solely fine-grained approach. This represents a potential limitation of this framework for brain-to-speech.

**Questions:**

Questions:
- Why was sex selected as the semantic class for the brain-to-speech case?
- Just confirming that: semantic features of guidance = course-grained features in the CLAP space? (might be helpful to state this more explicitly)

---

> ### Author Response · Authors · 2024-11-22
>
> We appreciate your questions and suggestions, which help improve the clarity and presentation of our work. Thank you for highlighting these points—we will make the necessary revisions to address your feedback.
>
> >Semantic analysis in the brain-to-speech case
>
> As the reviewer points out, the semantic decoding for speech performs poorly. Based on our results and experimental paradigm, we hypothesize that this may be because participants focus more on content in the speech condition. Compared to music and natural sounds, speech contains more information, and when the paradigm does not emphasize attending to the speaker, decoding becomes more challenging.
>
> However, we acknowledge that this explanation may not fully account for the results. It is also plausible, as you suggest, that the coarse-grained decoding framework may be less effective at capturing semantic quality from speech signals. This could stem from the inherent challenges of extracting meaningful high-level features from speech-related fMRI responses. We agree that further exploration is needed to investigate alternative approaches to improve the performance of semantic decoding for speech.
>
> >Why was sex selected as the semantic class for the brain-to-speech case?
>
> The semantics of speech are complex, encompassing features such as pitch, timbre, and emotion. We select sex as the semantic class primarily because it is one of the most straightforward and intuitive labels to annotate. Additionally, we draw inspiration from the semantic space constructed by the CLAP model. In the LAION-Audio-630K dataset, text captions like “A woman whispering softly” treat sex as a key component of semantic representation. Therefore, we consider sex decoding a reasonable task for extracting semantic information from brain signals. We have added the reasons for choosing sex in lines 418–421 of the updated draft.
>
> This choice is further supported by experimental evidence. As shown in Figure 5, using sex-labeled CLAP representations instead of poorly performing semantic representations significantly improves audio reconstruction accuracy. This demonstrates that sex effectively serves as a semantic prompt for guiding fMRI-based speech decoding.
>
> >semantic features of guidance = course-grained features in the CLAP space?
>
> Yes, the semantic features used for guidance are indeed the coarse-grained features in the CLAP space. Thank you for the suggestion—we will include an explicit statement to clarify this in the revised manuscript and improve readability.

---

### Official Review · Reviewer_Fjq3 · 2024-11-03

**Soundness:** 3
**Presentation:** 2
**Contribution:** 2
**Rating:** 5
**Confidence:** 3

**Summary:**

This paper presents a new method for reconstructing audio signals from fMRI responses to sounds. Specifically, it proposes a course-to-fine decoding strategy in which  fMRI responses from auditory cortex are decoded in a course-grained fashion into the semantic space of CLAP (which is not defined until page 3), and decoded in a fine-grained fashion into acoustic space.

I would not recommend this paper for acceptance. While the high-level approach seems interesting, the purpose of integrating neural data seems lacking in motivation, and the use of pre-trained representations in both training and evaluation seem to present a major confound.



Figure 5 is confusing. What are the ‘features’ that are input into the SVM? And given the results and discussion that there is ‘little semantic content in the semantic features’ (line 449) the initial claim that semantic prompts during decoding ‘[enhances] the quality of reconstructed audio’ seems perhaps exaggerated (lines 25-26).

A lot of the text is pushed to the appendix, making the official 10 pages lacking in sufficient detail and discussion.

A lot of the writeup feels pretty inside baseball to this reviewer (or perhaps this reviewer is just too far outside this particular game). e.g. CLAP is not explained (though there is a reference) until page 3.

**Strengths:**

Technically, the method introduced here is interesting and seems sophisticated.

The paper is well-written and makes use of highly relevant contemporary work—including studies done in machine learning and neuroscience—and uses this grasp of the literature to provide interpretations on different steps in their method.

The paper does a nice job of comparing many different methods in addition to their own.

**Weaknesses:**

1. Lack of Clear Motivation: This reviewer got stuck at square one: what is the goal of this work? The authors don't tell us. If the goal were to understand brain function this work would proceed differently, for example separately analyzing primary versus non primary auditory cortex, asking which brain regions are best modeled by each component of the model, comparing decoded signals to human behavioral data, etc. Alternatively, perhaps it could be intended for BCI applications. But what would those applications be? If you have access to a person's auditory cortex then you would already have access to the sound they were hearing, so there would be no point trying to decode that information from the brain. A paragraph in the appendix (A.7) attempts to provide a motivation, but it doesn't make sense to me. For example it mentions that this work could  "aid individuals with voice disorders". But if an individual had a voice disorder, one would want to generate the intended spoken information, not the heard information from auditory cortex.

It’s unclear how the findings here contribute to the ‘comprehension of the human auditory system’ as stated in the abstract (lines 27-29). While the coarse-to-fine method is inspired by the human auditory system, it is not quite a model of ‘each physiological structure of the auditory processing pathway’ (lines 96-97)

2. Possible confound.
The results on many of the metrics suggest that the direct decoding methods better reconstruct audio than the C2F-LDM proposed here (Table 1, 2). The novel C2F-LDM method does improve on measures of FD, FAD, KL, and CLAP, but these results seem to present a confound. The reconstruction makes use of many model features from pre-trained models and are subsequently evaluated on their similarity to pre-trained model features. Thus would higher scores not be unsurprising?

3. Is the fMRI data even relevant?
The paper should address the role of fMRI data and how it improves the methods described here. The majority of steps requires pre-trained models and predicting their ‘ground truth’ representations, but the optimal value of P=0.25 seems to suggest that using these ground truth representations without the neural data may improve reconstruction.

**Questions:**

Figure 5 is confusing. What are the ‘features’ that are input into the SVM? And given the results and discussion that there is ‘little semantic content in the semantic features’ (line 449) the initial claim that semantic prompts during decoding ‘[enhances] the quality of reconstructed audio’ seems perhaps exaggerated (lines 25-26).

A lot of the text is pushed to the appendix, making the official 10 pages lacking in sufficient detail and discussion.

A lot of the writeup feels pretty inside baseball to this reviewer (or perhaps this reviewer is just too far outside this particular game). e.g. CLAP is not explained (though there is a reference) until page 3.

---

> ### Author Response · Authors · 2024-11-22
> **Rebuttal by Authors [1/2]**
>
> Thank you for raising the insightful questions, which help us to better analyze and clarify an important aspect of our work. We truly appreciate your attention to detail and thoughtful engagement with our study.
>
> > Motivation
>
> Thank you to the reviewer for the important questions. We would like to clarify the motivation of this study:
>
> This paper focuses on technical breakthroughs in the brain-to-audio reconstruction task, with the primary aim of enhancing the accuracy of audio reconstruction from fMRI signals. This constitutes a crucial advancement in the development of auditory BCI technology. To achieve this, we propose a coarse-to-fine decoding approach inspired by neuroscience and demonstrate its effectiveness.
>
> > Applications
>
> We would like to clarify that our work focuses on induced BCI (where participants are presented with external stimuli), rather than spontaneous BCI, which involves decoding internally generated information, such as imagined speech or sound. The reviewer’s suggestion regarding generating intended spoken information aligns more with spontaneous BCI, which indeed presents greater challenges due to weaker task-relevant brain responses. In contrast, decoding auditory stimuli from the brain is a well-established research area with significant applications, as supported by numerous studies (Santoro et al., 2017; Bellier et al., 2023; Chen et al., 2024).
>
> A key application of reconstructing auditory stimuli lies in auditory attention decoding (AAD), where the goal is to reconstruct and enhance the sound source a listener is focusing on in a multi-speaker environment (Van Eyndhoven et al., 2016; O’Sullivan et al., 2017). This has practical uses in hearing aid enhancement and communication assistance in noisy environments .
>
> Furthermore, while our current work is focused on reconstructing perceived audio, it can serve as a foundation for future research on reconstructing imagined audio. Previous studies in vision (Horikawa & Kamitani, 2017) and audition (Tang et al., 2023) have shown that decoders trained on perception tasks can generalize to imagination tasks, providing a pathway for extending this line of research to spontaneous BCI.
>
> We acknowledge that some parts of the introduction may have caused misunderstanding regarding the motivation and applications of our work. We have included a more detailed introduction to the applications of reconstructing auditory stimuli in lines 40–44 of the updated draft.
>
> >Contributions to neuroscience
>
> First, as illustrated in Figure 1, our work highlights the correspondence between the components of our model and the anatomical structures of the auditory system. This mapping helps to elucidate the roles of different brain regions in auditory processing.
>
> Second, our approach not only draws inspiration from established findings in neuroscience but also uses engineering practice to validate these findings. For instance, the hierarchical processing characteristics modeled in our coarse-to-fine decoding framework align with known properties of the auditory pathway, reinforcing the plausibility of these mechanisms. This bidirectional validation between computer science and neuroscience provides valuable insights into the human auditory system and contributes to advancing our understanding of its processing mechanisms.
>
> > Pretraining in training and evaluation
>
> Thank you for raising this important point. We would like to clarify the following:
>
> **Choice of Metrics**: The metrics FD, FAD, KL, and CLAP Score are widely recognized in the audio domain as standard measures for evaluating audio quality. These metrics are commonly used, enabling fair and meaningful comparisons across different methods.
>
> **Pretrained Models in Our Framework**: The pretrained models (CLAP, AudioMAE, LDM) used in our method serve specific functional roles, such as auditory feature extraction, neural decoding at different levels, and audio generation. Their selection is driven by functional requirements, not evaluation considerations.
>
> **Evaluation Bias**: Among the evaluation models (PANNs, VGGish, CLAP), only CLAP overlaps with those used during training. However, all metrics, including those unrelated to CLAP, consistently demonstrate improvements. This consistency indicates that the observed gains reflect genuine enhancements from our method rather than evaluation bias.
>
> We believe these points address the potential confound and validate the improvements achieved by our approach.

---

> ### Author Response · Authors · 2024-11-22
> **Rebuttal by Authors [2/2]**
>
> > Roles of fMRI and the ground truth
>
> The value of $P=0.25$ indicates that during training, there is a 75% probability of using brain-decoded semantic features and a 25% probability of using the ground truth semantic features. This setup reflects a moderate guidance from the ground truth, rather than replacing the fMRI data. The reconstruction still primarily relies on features decoded from the fMRI data. This approach helps reduce the impact of decoding noise and improve the stability of the reconstruction by bringing the decoded space closer to the original audio feature space. Additionally, it leaves room for semantic prompts, enabling conditional reconstruction without retraining.
>
> The value of $P$ represents a trade-off between decoding from fMRI and using the ground truth, and it is not the case that higher values always lead to better results. Since fMRI data is used exclusively during testing, higher $P$ values would create a mismatch between training and testing conditions, which negatively impacts reconstruction accuracy. For instance, in the Brain2Music dataset, reconstruction quality at $P = 0.5$ is significantly worse than at $P = 0.25$. Therefore, $P = 0.25$ is the optimal value determined based on multiple evaluation metrics, and fMRI data remains the most critical source of information for reconstruction.
>
> We have clarified the meaning and selection of the $P$ value in lines 476-480 of the updated draft.
>
> >Input into the SVM and conflict between line 449 and line 25
>
> To address these questions, we outline the logic of Sections 3.3 and 3.4:
>
> In Section 3.3, we investigate whether the improved reconstruction accuracy from coarse-to-fine decoding is due to enhanced semantic content in the decoded acoustic features. To test this, we input the **decoded acoustic features** from **coarse-to-fine decoding and fine-grained decoding** into an **SVM** classifier to measure their semantic information.
>
> The results show that semantic enhancement in decoded acoustic features is not observed in all cases, as it depends on the quality of coarse-grained semantic decoding. For example, in the **Brain2Speech dataset**, where semantic decoding performs poorly, **"there is little semantic content in the semantic features" (line 449)**. It leads to reduced semantic content in the decoded acoustic features for some subjects (Figure 5(b)).
>
> Building on these findings, Section 3.4 explores how external semantic prompts can address suboptimal semantic decoding. As shown in Table 3, we demonstrate that **semantic prompts** effectively improve reconstruction quality in the **Brain2Speech dataset**, which supports our claim in the abstract: "by employing semantic prompts during decoding, we enhance the quality of reconstructed audio **when semantic features are suboptimal**" **(lines 25–26)**.
>
> Overall, our experiments are connected: Brain2Music and Brain2Speech datasets, with their differing semantic decoding performance, jointly illustrate the role of coarse-to-fine decoding and highlight how semantic prompts improve reconstruction quality.
>
> >Appendix and main text
>
> Thank you for your feedback. We understand the concern regarding the level of detail and discussion in the main text. Due to the strict page limit, we have prioritized presenting the core ideas and results in the main body, while placing some supplementary details in the appendix to ensure clarity and conciseness. Although the main text is currently at full capacity, we will make an effort to further enrich the discussion within the existing space to provide more depth and context where possible.
>
> >Explanation of terms
>
> Thank you for pointing this out. We have revised the introduction in lines 103-106 to provide a brief explanation of key terms and concepts, such as CLAP, to improve clarity and ensure that the paper is more approachable for a broader audience.
>
>
>
> **Additional references:**
>
> - Van Eyndhoven, Simon, Tom Francart, and Alexander Bertrand. "EEG-informed attended speaker extraction from recorded speech mixtures with application in neuro-steered hearing prostheses." *IEEE Transactions on Biomedical Engineering* 64.5 (2016): 1045-1056.
> - O’Sullivan, James, et al. "Neural decoding of attentional selection in multi-speaker environments without access to clean sources." *Journal of neural engineering* 14.5 (2017): 056001.
> - Horikawa, Tomoyasu, and Yukiyasu Kamitani. "Generic decoding of seen and imagined objects using hierarchical visual features." *Nature communications* 8.1 (2017): 15037.
> - Tang, Jerry, et al. "Semantic reconstruction of continuous language from non-invasive brain recordings." *Nature Neuroscience* 26.5 (2023): 858-866.

---

> > ### Comment · Reviewer_Fjq3 · 2024-11-26
> > **Reply by Reviewer Fjq3**
> >
> > Thank you for your detailed response. The motivation behind this study is clearer, as are its applications to fields like auditory attention decoding. However, my initial question still remains—in which scenario would one have fMRI data but not the ground truth audio? Or is this model only to use fMRI data during some offline training period?
> >
> > The motivation and utility of pre-trained model features in your method still also confuses me. The different components of your framework are inspired by the brain’s hierarchy, but you also claim that they ‘provide valuable insights into the human auditory system’ and reveal something of its ‘processing mechanisms.’ This reasoning seems a bit circular to me.
> >
> > Recent studies also suggest that task optimization leads to convergent representations in models (Huh et al, 2024 arXiv). So while there is not complete overlap between the features used in your method and those used in the evaluation metrics, perhaps there is still some confounding similarity? This would have to be empirically tested within your framework.
> >
> > Thank you for the clarified explanation of sections 3.3 and 3.4. The results however still seem misaligned with the claim of the paper: that ‘coarse-to-fine decoding is superior to solely fine-grained decoding.’
> >
> > Thank you for the clarification of terms in the main text.
> >
> > After reconsideration and due to concerns outlined above, this reviewer has decided to keep their original score.

---

### Official Review · Reviewer_JVh6 · 2024-11-04

**Soundness:** 2
**Presentation:** 1
**Contribution:** 2
**Rating:** 3
**Confidence:** 5

**Summary:**

This work designs a system to reconstruct audio signals from human fMRI data collected while listening to natural sounds. The system separately handles fine-grained information (i.e., the precise timing and frequencies of the sounds) and course-grained information (semantic information such as the class of sound) by using different neural network features for each branch of the architecture. The authors evaluate their method on three different fMRI datasets and multiple different evaluation methods capturing the fine-grained and semantic nature of the sounds.

**Strengths:**

The paper tackles an interesting and challenging problem of trying to decode auditory fMRI activity. I like how it ties back into classic ideas of course-to-fine reconstruction. It also takes advantage of many recently proposed auditory models, similarity metrics, and auditory datasets.

**Weaknesses:**

* The paper combines many different pre-trained systems. This makes the methods used difficult to understand, but also makes it hard to evaluate where things might be going wrong, and which pieces are critical. Due to the unknown inherit biases of each system, it is difficult to see how the work could actually be useful for understanding the human auditory system.

* As currently written, the presentation of results is confusing. The text should point to each set of results when they are discussed, rather than stating (line 357-359) that qualitative results are displayed in Tables 1 and 2. Additionally, it would be helpful if the columns were in some way separated or labeled for the “fine grained” vs. “semantic” measures.

**Questions:**

1)	For each of the metrics, it would be helpful to understand what a good value is, and if these are getting at all close to that value. Currently, it is difficult to interpret whether the observed differences are at all significant. For instance, for the semantic representations one could use a baseline score that is two different samples in time from the same audio clip.  And maybe a PCC and PSNR score could be referenced to samples with additive Gaussian noise of various SNR levels at the waveform?

2)	Listening to the supplement audio files, I am not so sure that the proposed method is actually doing something reasonable. The samples “sound” more natural (which is expected from including a diffusion model in the pipeline), but they are often completely different sounds than the initialized audio, essentially hallucinations. PSNR levels of ~15-18 seem like they might be noise, relative to the original sound.

3)	In the decoding section, one of the experiments is on male/female decoding which is called a “semantic” task. However, I believe a decent amount of male/female decoding can be achieved from simply looking at the overall power spectrum of the sound (male voices are generally lower than female voices). Thus, this doesn’t seem particularly “semantic”.

4)	The metrics are not defined enough in the main text (Lines 311-319). At a minimum, the acronyms need to be defined. For instance, what is “PCC” and “PSNR”? Although some audio researchers may be familiar with these measurements, a more general ICLR attendee may not.

5)	How many repetitions of each sound are present in each dataset and are these averaged for the analysis? More generally, how is fMRI measurement noise taken into account for the analysis? I.e., is there a sort of “noise ceiling” defined on the reconstruction?

Somewhat minor suggestions for specific lines:
* Line 075-077: This sentence doesn’t make sense to me. How does the “high-dimensionality” play a role here? The main challenge of fine-grained decoding is the lack of resolution in time and the inherently noisy signal of the fMRI response.
* Lines 086-088: These references seem out of place, as the decomposition of sound into difference frequencies in the cochlea dates much further back than these papers. Some good references might be work by Shihab Shamma (maybe https://ieeexplore.ieee.org/abstract/document/119739, or https://pubmed.ncbi.nlm.nih.gov/16158645/) , but I would encourage the authors to perhaps cite classic textbooks or review articles on auditory processing
* Lines 147-152: Which features are used in CLAP? The final output features or an intermediate stage? This should be mentioned in this section.
* Line 231: “unpatchify” is not defined.

---

> ### Author Response · Authors · 2024-11-22
> **Rebuttal by Authors [1/3]**
>
> Thank you to the reviewer for all the questions and suggestions, which have prompted much reflection on my part. We will address the expression issues mentioned in the updated version.
>
> > The paper combines many different pre-trained systems.
>
> We understand this concern. Regarding the use of multiple pre-trained systems, we would like to clarify the following points:
>
> Firstly, neural decoding involves a complex mapping from high-dimensional brain signals to high-dimensional perceptual information. Decomposing the task into multiple sub-modules, each focusing on a specific function, makes the system more controllable and easier to understand. The use of pretrained models allows us to leverage existing domain knowledge (introduced in lines 103–106 of the updated draft), avoiding the challenges of training from scratch.
>
> Secondly, we have demonstrated the necessity of each module through systematic ablation study:
>
> - The comparison of C2F-LDM with Fine-LDM highlights the importance of coarse-grained decoding.
> - The comparison of C2F-LDM with C2F-Decoder showcases the advantages of LDM over directly using the AudioMAE Decoder.
> - The comparison with direct decoding methods proves that the framework separating neural decoding and generation can achieve reasonable signal reconstruction accuracy while maintaining high perceptual quality.
>
> Finally, our module design aligns with the physiological structure of the auditory processing pathway. This design not only aids in understanding the functions of different brain regions in auditory processing but also facilitates the validation of the hierarchical processing characteristics of the auditory system, contributing to the understanding of auditory mechanisms.
>
> > Presentation of results
>
> Thank you for your suggestions. In lines 354–357 of the updated draft, we have introduced the three sections of the tables, which are analyzed sequentially in the following discussion, making it easier for readers to follow our analytical process. Revisions have also been made in lines 362–368 and 371–376. Regarding the table structure, the first two columns represent low-level fidelity quality, while the last four columns reflect high-level perceptual quality. This distinction is clarified in lines 312–313 of the updated draft, helping readers better understand the metrics and results.
>
> >Upper bound and baseline
>
> Thank you for your suggestions. We have included an upper bound and additional baselines with different SNR levels to evaluate our method.
>
> Upper Bound: We use the reconstruction results conditioned on the ground truth acoustic feature $c_{gt}$ as the upper bound, which reflects the theoretical maximum achievable under the current architecture.
>
> Baseline: We randomly sample audio from the training set and introduce Gaussian noise at SNR levels of 10dB, 15dB, and 20dB. These SNR levels reflect typical distortions observed in neural reconstruction, simulating reconstructed audio across different quality levels.
>
> The results are as follows:
>
> Brain2Sound
>
> |             | **PCC↑**  | **PSNR↑**  |  **FD↓**   | **FAD↓**  |  **KL↓**  | **CLAP↑** |
> | :---------: | :-------: | :--------: | :--------: | :-------: | :-------: | :-------: |
> | 10dB noise  |   0.244   |   14.435   |   67.783   |  13.089   |   4.799   |   0.191   |
> | 15dB noise  |   0.280   |   14.581   |   64.769   |  12.086   |   4.882   |   0.201   |
> | 20dB noise  |   0.307   |   14.573   |   62.289   |  11.906   |   4.946   |   0.204   |
> |  Fine-LDM   |   0.376   |   14.624   |   49.827   |  10.803   |   2.895   |   0.265   |
> | **C2F-LDM** | **0.418** | **15.103** | **44.003** | **9.324** | **2.697** | **0.275** |
> |    *Upp*    |  *0.934*  |  *26.349*  |  *9.872*   |  *3.502*  |  *0.224*  |  *0.300*  |
>
> Brain2Music
>
> |             | **PCC↑**  | **PSNR↑**  |  **FD↓**  | **FAD↓**  |  **KL↓**  | **CLAP↑** |
> | :---------: | :-------: | :--------: | :-------: | :-------: | :-------: | :-------: |
> | 10dB noise  |   0.380   |   16.990   |  22.751   |   8.412   |   0.780   |   0.406   |
> | 15dB noise  |   0.410   | **17.266** |  12.835   |   4.511   |   0.667   |   0.441   |
> | 20dB noise  |   0.428   |   17.227   |   7.147   |   2.096   |   0.620   |   0.466   |
> |  Fine-LDM   |   0.419   |   15.526   |   6.412   | **1.273** |   0.548   |   0.512   |
> | **C2F-LDM** | **0.454** |   15.883   | **6.102** |   1.504   | **0.520** | **0.530** |
> |    *Upp*    |  *0.922*  |  *26.459*  |  *2.204*  |  *0.547*  |  *0.081*  |  *0.888*  |

---

> ### Author Response · Authors · 2024-11-22
> **Rebuttal by Authors [2/3]**
>
> Brain2Speech
>
> |             | **PCC↑**  | **PSNR↑**  |  **FD↓**  | **FAD↓**  |  **KL↓**  | **CLAP↑** |
> | :---------: | :-------: | :--------: | :-------: | :-------: | :-------: | :-------: |
> | 10dB noise  |   0.236   |   13.876   |  44.531   |  20.563   |   1.116   |   0.336   |
> | 15dB noise  |   0.260   |   14.541   |  29.539   |  16.851   |   0.868   |   0.378   |
> | 20dB noise  |   0.279   |   14.911   |  18.278   |  13.394   |   0.662   |   0.419   |
> |  Fine-LDM   |   0.357   |   14.385   |  12.706   |   4.820   |   0.885   |   0.420   |
> | **C2F-LDM** | **0.393** | **15.260** | **9.726** | **4.623** | **0.616** | **0.471** |
> |    *Upp*    |  *0.967*  |  *28.192*  |  *1.201*  |  *1.452*  |  *0.035*  |  *0.886*  |
>
> The results indicate that in the Brain2Sound dataset, the rich diversity of audio leads to significant distribution differences between the training and testing sets, resulting in suboptimal performance of the baseline model. In contrast, our method shows a marked enhancement. For the Brain2Music and Brain2Speech datasets, where the distribution differences between the training and testing sets are minimal, the baseline performs relatively well on certain metrics. Nevertheless, in these domain-specific tasks, our method still maintains a good distribution match and demonstrates advantages in terms of fidelity.
>
> > Similarity of samples
>
> We acknowledge that a lower PSNR may indicate a significant discrepancy between the reconstructed audio and the original signal. This is primarily due to the limitations of fMRI, which has low temporal resolution and SNR, making it challenging to recover the spectrograms accurately. By incorporating diffusion models, our approach aims to improve the naturalness and details of the generated audio. However, it also relies on the model's generative capabilities to fill in missing information, which can lead to differences between the reconstructed and original audio.
>
> However, the main contribution of our work lies in the significant progress in the brain-to-audio task compared to existing brain decoding methods. No model so far has reconstructed audio that sounds highly similar to the original. We validate our improvements in reconstruction similarity through comprehensive quantitative metrics and various analyses, including decoding accuracy (Figure 8), semantic decoding (Figure 5), and semantic prompts (Table 3). These findings highlight the contributions of our method.
>
> We also recognize areas for improvement in balancing the naturalness and consistency of the generated audio. Future directions include optimizing the conditions of diffusion model to reduce "hallucinations", introducing more semantic prompts to enhance fMRI decoding, and integrating brain signals with high temporal resolution (such as EEG and MEG) to improve temporal consistency and fidelity. These advancements could further enhance the quality of reconstructed audio.
>
> > Gender and semantics
>
> We understand that the spectrogram differences between male and female voices in audio signals can indeed provide an important basis for distinguishing gender. However, this does not imply that gender information is solely low-level spectrogram features. The brain’s perception of gender relies not only on spectral characteristics processed in the cochlea but also on the integration of various features, such as pitch, timbre, and emotion, within higher auditory processing areas.
>
> In the context of semantic modeling, we can reference the semantic space constructed by the CLAP model. In the LAION-Audio-630K dataset, text captions may be as brief as "A woman whispering softly," which indicates that gender is treated as a part of the semantic representation. Therefore, we believe that gender decoding can be a reasonable task for semantic decoding from brain signals.
>
> Our experiments aim to explore whether gender information can be decoded from brain signals to assess the effectiveness of semantic decoding. Gender is chosen because it is an intuitive and easily annotated label. More importantly, our results show that gender serves as an effective semantic prompt for guiding fMRI decoding. In Figure 5, replacing a poorly performing semantic representation with CLAP representation of gender significantly improves the accuracy of audio reconstruction.
>
> We acknowledge the lack of explanation here. We have added the reasons for choosing gender in lines 418–421 of the updated draft.
>
> > Introduction to metrics
>
> Thank you for your suggestion. We have enriched the introduction of metrics in the updated draft.

---

> ### Author Response · Authors · 2024-11-22
> **Rebuttal by Authors [3/3]**
>
> > Noise ceiling
>
> Repeated samples in the datasets
>
> - In the Brain2Music and Brain2Speech datasets, each audio sample is unique with no repetitions.
> - In the Brain2Sound dataset, each audio sample has multiple repetitions (see Section A.1 for details). During testing, we averaged the repeated fMRI data to enhance the signal-to-noise ratio. This method is consistent with the approach used by Park et al., facilitating comparison of results.
>
> Definition of noise ceiling
>
> In neural encoding tasks, the noise ceiling is typically defined by calculating the correlation between trials. However, in our method, reconstruction is performed individually for each sample, and therefore, we do not adopt this approach. Instead, we design a reconstruction upper bound, which is explained in detail above.
>
> > Line 075-077
>
> Low temporal resolution and signal-to-noise ratio are indeed major challenges in fMRI decoding. However, in this context, fine-grained decoding is primarily being compared to the low-dimensional coarse-grained decoding mentioned in line 078. High-dimensional data implies more information that is coupled, making it harder to decode high-dimensional features than low-dimensional features.
>
> We conduct a comparative experiment: using ridge regression, the decoding PCC for CLAP embeddings is significantly higher than that for AudioMAE embeddings, as shown in the tables below. This demonstrates that decoding high-dimensional features is more challenging.
>
> |          |  S1   |  S2   |  S3   |  S4   |  S5   | sub-001 | sub-002 | sub-003 | sub-004 | sub-005 | UTS01 | UTS02 | UTS03 | UTS05 | UTS06 | UTS07 | UTS08 |
> | :------: | :---: | :---: | :---: | :---: | :---: | :-----: | :-----: | :-----: | :-----: | :-----: | :---: | :---: | :---: | :---: | :---: | :---: | :---: |
> |   CLAP   | 0.547 | 0.571 | 0.571 | 0.557 | 0.571 |  0.221  |  0.245  |  0.257  |  0.231  |  0.247  | 0.245 | 0.253 | 0.275 | 0.203 | 0.291 | 0.254 | 0.230 |
> | AudioMAE | 0.180 | 0.188 | 0.174 | 0.163 | 0.201 |  0.045  |  0.049  |  0.058  |  0.035  |  0.055  | 0.071 | 0.075 | 0.086 | 0.058 | 0.082 | 0.071 | 0.056 |
>
> > Lines 086-088
>
> Thank you to the reviewer for providing the references and reading directions. We cite the following articles and textbooks:
>
> - Pickles, J. O. "An introduction to the physiology of hearing." (1988).
> - Shamma, Shihab A., and Christophe Micheyl. "Behind the scenes of auditory perception." *Current opinion in neurobiology* 20.3 (2010): 361-366.
> - Schnupp, J. *Auditory Neuroscience: Making Sense of Sound*. MIT Press, 2011.
> - Moore, Brian CJ. *An introduction to the psychology of hearing*. Brill, 2012.
>
> > Lines 147-152
>
> Thank you for your suggestions. We use the output from the final layer of CLAP, and we have included this information in line 155 of the updated draft.
>
> >Line 231
>
> "Unpatchify" is the reverse process of "patchify", which reconstructs the original spectrogram by reassembling all the patches in their original order.

---

> > ### Comment · Reviewer_JVh6 · 2024-11-26
> > **Response to author rebuttal**
> >
> > Thank you to the authors for the responses. The presentation of the results has improved, and I think the additional references seem more appropriate.
> >
> > However, I remain puzzled by the author's motivation for the work and the system design (this concern also seems to be noted by other reviewers). If the end goal is just engineering purposes to reconstruct neural activity, this type of approach may be fine and indeed lead to good performance, but I would argue that it actually makes it *harder* to understand compared to a simple model. This understanding is critical when it comes to actually using the model to learn something about the human auditory system. The audio examples are quite confusing for this reason, as I noted they sound like there are clear hallucinations, and as such this doesn’t actually seem to be *reconstructing* the audio signal, but rather generating a sound in a complex, uninterpretable way. For these reasons, in its current form, the paper doesn’t clearly contribute substantial progress to either the pure engineering problem of reconstruction or to the scientific question of studying the human auditory system. The authors’ responses to my concerns about these things left me more perplexed than satisfied.
> >
> > I unfortunately don’t think this motivational concern can be addressed without substantial revisions that are beyond the scope of the rebuttal period, so I hope the authors take time to refine the paper based on the reviewer comments. There is something quite interesting in this paper, but in its current form it is too difficult to know what to take away from it, and my score stays the same.
> >
> > One more note:
> >
> > >Upper bound and baseline
> >
> > These results are confusing. I would expect that the PSNR measured between the original audio and the audio + Gaussian noise would go down with increasing noise levels. However, it seems to increase in the table (in a non-monotonic fashion, which is even more puzzling).
> >
> > Additionally, I don’t see strong support from these tables for the claim that the method performs well on “certain metrics”. Which metrics is it performing well on? It generally seems to have scores that are quite far from the upper bound, and are much closer to the baseline values (possibly non-significant differences, as things like error bars are not given). In a future submission, I would encourage the authors to think carefully about the comparisons to make for this analysis so that “good” performance is well-defined.

---

### Official Review · Reviewer_DqUq · 2024-11-04

**Soundness:** 3
**Presentation:** 1
**Contribution:** 3
**Rating:** 5
**Confidence:** 3

**Summary:**

This paper aims to achieve audio reconstruction from an fMRI brain signal via a coarse-to-fine approach. The idea is to replicate the audio processing stream in the human auditory cortex. The method is threefold: first, it uses a CLIP-based approach to extract an audio representation (low-level description) from the fMRI data signal. From these initial features, a high-dimensional description of the auditory feature is obtained via a guided AudioMAE. Finally, these high-level features are used as a condition of a Latent Diffusion model for mel-spectrogram reconstruction and, thus, audio reconstruction. The method achieves high results on three publicly available datasets, demonstrating strong performance for both low-level and high-level audio metric reconstructions, showing improvement compared to the direct reconstruction of mel-spectrogram and other high-level approaches which omit low-level features.

**Strengths:**

- The technical contributions of the paper are strong. The methodology, with the threefold approach, is well thought out and carefully implemented. Many implementation details are provided, and the notations are clear and easy to follow. This is a great point.

- Figure 2 is clear and provides sufficient details to understand the intricate methodology

- The new method is compared against various baselines and three openly available datasets. Also, the code is made public, and additional results are provided. This is a great point for the openness of science.

- Various ablation studies are done to confirm the contributions of specific modules (for instance the diffusion reconstruction vs using the MAE decoder).

**Weaknesses:**

- One weakness of the paper is the limited neuroscientific motivations for audio reconstruction as a way to understand auditory mechanisms (first paragraph of the introduction). It is not clear from the introduction or the conclusion how solving this task could help to understand better the auditory processes (line 28  "This research contributes to the comprehension of the human auditory
system" ). Are these models generalisable between subjects? can a model trained on a single subject be used for another subject? can they be used to identify specific features related to language disorders?

- Another major weakness in this study is the lack of clarity in the writing, which prevents us from understanding some of the motivations and results/discussion. The characterisation of "high-level/low-level" features and "coarse/fine-grained" features throughout the paper is very unclear. Some of these terms seem interchangeable in how they are employed; the paper would greatly benefit from a clear definition (see more related questions in the next section).

- Some points of the methodology lack details and/or seem to diminish the results (see questions in the next section). For instance, the modelling of the brain signal with ridge regression, which overlooks the spatial autocorrelation of the brain signal and structure of the auditory cortices, is not motivated. Sections 3.3 and 3.4 lack clarity, it is not clear what they aim to achieve with respect to the general motivation of the paper, and the conclusion are not clear and even seem to diminish previous results (for instance Figure 5 and  line 457 to 460) - see more questions below.

- There are some important missing descriptions regarding the dataset and the training procedure. How is the data split between training and testing? is it subject-wise or dataset-wise? Are the same subjects used both for training and testing? functional alignment between subjects if direct comparison between them?

**Questions:**

**Improving clarity and motivations**

- The distinction between low-level (coarse) and high-level (fine-grained) features is confusing, lacks clarity and contributes negatively to the overall appreciation of the paper. Could the authors precise what they refer to as high-level or fine-grained features and as coarse or low-level features? It is confusing to refer to "semantics" as coarse, as they are supposed to provide very precise and high-level descriptive information. Similarly, the spectrogram is often referred to as high-level, although it describes low-level features similar to the features extracted by the cochlea (line 86).

- The description of the CLAP training (with aligned textual descriptions and audio) (line 147) and the use of prompts for incorporating music genres and phenotype information  (line 117), are considered as coarse features, why?

- The paper refers to as "inverse pathway of auditory processing" (line 17). If it is inverse; shouldn't it go from high-level to low-level? seems that the method goes from coarse-to-fine ... "AudioMAE latent feature as the fine-grained acoustic embedding of audio" (line 182) isn't it coarse information?

- Typo line 466 ->"it's better"

- In line 52, what are the DNN features referred to? could the authors provide more explanations?


**Methodology/Results:**

-  It is unclear how the Semantic Decoder extracts semantically rich information (line 137); is it validated somehow?

- One of the baseline modes is a Transformer that goes from voxel space to mel-spectrogram space. Could the authors provide more information about this implementation, as it does not seem straightforward?

- Could the authors provide some motivations for using ridge regression to model brain signals (referred to as $x$ in the paper)? does this consider the autocorrelation of brain cortical activation and using the structure and spatial organisation of the auditory cortex?

- Regarding the generation process and the use of the latent diffusion model, how much do the authors consider providing the ground truth as input to help the model (figure 3)? Isn't the conditioning should be enough?

- In tables 1, 2, and 3, are the results averaged across subjects? if yes could you provide the std?

- In Table 1, why is C2F-LDM low for PCC and PSNR?

- The objective of section 3.3. and 3.4 are not very clear, for instance, lines 457 to 460. More importantly, the results do not seem to go in the direction of the conclusion of the paper, e.g. in Figure 5. b. why does the fine-grained decoding perform better for almost all experiments? The hypothesis made in section 3.3, e.g. "This could be attributed to participants being more focused on the content of the stories during fMRI signal collection, potentially disregarding the speaking style of the speaker." (line 453) is used to motivate the next section 3.4 but is relatively weak and unclear. In section 3.4, The Brain2Music dataset performs better without prompts, although the prompts are supposed to incorporate important information such as the music genre. Why use prompts, then? seems somehow superficial.

**Data**:

- what about variation in performance across subjects? are the models trained per sujects? or models are used across subjects?

- how are the voxels in Table 4 extracted? Is it from a template? subject-based parcellation?

- Are the datasets aligned functionally (on top of anatomical registration) so that similar voxels and auditory regions across subjects can be compared?

- For all datasets, the audio clips seem to be between 1.5 and 4 seconds. Is it enough to aim at reconstructing audio from a very short fMRI temporal window? What about using larger windows?

---

> ### Author Response · Authors · 2024-11-22
> **Rebuttal by Authors [1/3]**
>
> We would like to express our gratitude to the reviewer for the efforts in reviewing our manuscript. We will address the deficiencies in definitions and the inadequacies in our statements as noted in the review.
>
> In response to the weaknesses and issues highlighted by the reviewer, we have summarized them (weaknesses 1-4, questions 1.1-1.5, 2.1-2.7, and 3.1-3.4) as detailed below. We hope to continue our dialogue with the reviewer regarding these points.
>
> > Motivation [weakness1]
>
> The focus of our paper is on the AI task of Brain2Audio, aimed at improving reconstruction accuracy. To achieve this goal (and to enhance the interpretability of the model), we design a coarse-to-fine decoding approach based on insights from neuroscience and demonstrate its effectiveness.
>
> Furthermore, understanding auditory mechanisms is an additional contribution of our work. First, we illustrate the correspondence between various components of the model and the anatomical structure of the auditory system in Figure 1, which aids in comprehending the functions of different brain regions in auditory processing. Second, our work not only draws upon findings from neuroscience (lines 85-95) but also validates the hierarchical processing characteristics of the auditory system through engineering practice. This bidirectional validation between computer science and neuroscience holds value for understanding the auditory system.
>
> > Definitions of high/low-level and fine/coarse-grained [weakness2 question1.1-1.3]
>
> "Coarse-grained" refers to features that contain partial information about the audio, which in our paper pertains to high-level features, namely semantic features. In contrast, "fine-grained" refers to features that encompass all information from the audio, including both high-level and low-level features (such as acoustic details, like rhythm, pauses, and so on).
>
> Therefore, semantic features are considered coarse because they do not include the entirety of the original audio information. They only provide part of the high-level information, such as the descriptive text in the training of CLAP and the simple prompts we provide [question 1.2].
>
> In comparison, AudioMAE is trained on a generative task that retains both high-level and low-level information. Hence, we refer to the latent features of AudioMAE as fine-grained features [question 1.3].
>
> > Typo [question 1.4]
>
> Thank you for pointing this out. Our expression is not sufficiently formal. We have made revisions in the updated draft.
>
> > What are the DNN features referred to? [question 1.5]
>
> DNNs refer to various deep acoustic models or multimodal models. DNN features include intermediate representations from discriminative models, such as VGGish-ish (Iashin & Rahtu, 2021) used in Park et al. (2023), as well as latent representations from autoencoders used in Chen et al. (2024). Additionally, it encompasses features from multimodal models, such as MuLan (Huang et al., 2022b) used in Denk et al. (2023).
>
> > Validation of semantic decoding [question 2.1]
>
> Regarding the decoding performance, there are differences across the various datasets. The regression PCCs for the three datasets are 0.563±0.009, 0.240±0.012, and 0.250±0.026, respectively.
>
> In Section 3.3, we discuss the effectiveness of semantic decoding and suggest that it affects the semantic representation of fine-grained features. Specifically, in Figure 5, we classify the decoded semantic features for two datasets, with the average accuracy being 0.257 (chance level = 0.1) and 0.528 (chance level = 0.5), respectively. Additionally, Figures 9 and 10 provide visualizations, which indicate that the Brain2Music decoding performs better. These evaluations contribute to the assessment of semantic decoding.
>
> > Implementation of the Transformer Baseline [question 2.2]
>
> First, an fMRI token is obtained through voxel selection (line 210). Then, the token is encoded using a Transformer Encoder, followed by a projection to the mel-spectrogram through a linear layer. This baseline is implemented primarily to test existing reconstruction methods and does not fully leverage the sequence modeling capabilities of the Transformer, which presents certain limitations.

---

> ### Author Response · Authors · 2024-11-22
> **Rebuttal by Authors [2/3]**
>
> > Ridge regression [question 2.3 weakness 3]
>
> The reasons for choosing ridge regression include its simplicity, training stability, and computational efficiency. Given the high dimensionality of brain signals and the relatively small sample size, the L2 regularization of ridge regression effectively prevents overfitting, and it has been widely applied and validated in fMRI decoding research. For instance, Naselaris et al. (2011) explain why linear models (including ridge regression) perform well in fMRI decoding, while Varoquaux et al. (2017) confirm the advantages of ridge regression in terms of stability and generalization. Pasley et al. (2012) are among the first to use ridge regression to reconstruct speech from auditory cortex activity, and Yang et al. (2015), Hassan et al. (2018), and Bellier et al. (2023) also employ ridge regression (linear regression) as a decoding method.
>
> We do not explicitly model the spatial structure of the auditory cortex; instead, we implicitly consider spatial organization using voxel selection by selecting the voxels with the highest responses. Future work could explore more complex modeling methods.
>
> >Input of the ground truth for LDM [question 2.4]
>
> This is primarily intended to establish a reconstruction upper bound using ground truth as a condition for model evaluation. The improvement in reconstruction performance is not significant, as features decoded from fMRI can be used as the sole condition during reconstruction.
>
> >Experimental setup and standard deviation [question 2.5 3.1 weakness 1 4]
>
> In our experiments, each subject is trained and tested individually, and the metrics are averaged across subjects. We have expressed this more clearly in lines 318–319 of the updated draft. Currently, cross-subject usage is not supported. The standard deviation is as follows, and the overall performance appears to be relatively normal.
>
> Brain2Sound
>
> |             |  **PCC↑**   |  **PSNR↑**   |    **FD↓**    |   **FAD↓**   |   **KL↓**   |  **CLAP↑**  |
> | :---------: | :---------: | :----------: | :-----------: | :----------: | :---------: | :---------: |
> |     LiR     | 0.607±0.002 | 17.506±0.162 | 105.113±1.258 | 40.877±0.307 | 4.027±0.034 | 0.175±0.002 |
> |     MLP     | 0.566±0.006 | 17.310±0.169 | 98.358±1.520  | 38.045±0.431 | 4.020±0.027 | 0.164±0.003 |
> |   BiLSTM    | 0.580±0.005 | 17.381±0.151 | 112.031±1.004 | 39.895±0.561 | 3.948±0.028 | 0.180±0.002 |
> | Transformer | 0.581±0.000 | 17.676±0.093 | 104.118±1.344 | 39.484±0.888 | 3.764±0.033 | 0.177±0.002 |
> | Park et al. | 0.394±0.006 | 15.406±0.087 | 88.456±0.947  | 12.694±0.496 | 2.251±0.116 | 0.268±0.003 |
> |  Fine-LDM   | 0.376±0.023 | 14.624±0.102 | 49.827±2.450  | 10.803±0.796 | 2.895±0.064 | 0.265±0.009 |
> | C2F-Decoder | 0.595±0.010 | 17.385±0.515 | 95.565±4.964  | 35.775±1.763 | 3.748±0.131 | 0.179±0.009 |
> |   C2F-LDM   | 0.418±0.008 | 15.103±0.313 | 44.003±1.564  | 9.324±0.878  | 2.697±0.079 | 0.275±0.007 |
>
> Brain2Music
>
> |             |  **PCC↑**   |  **PSNR↑**   |   **FD↓**    |   **FAD↓**   |   **KL↓**   |  **CLAP↑**  |
> | :---------: | :---------: | :----------: | :----------: | :----------: | :---------: | :---------: |
> |     LiR     | 0.637±0.001 | 19.353±0.055 | 47.710±0.309 | 18.247±0.246 | 0.997±0.039 | 0.223±0.002 |
> |     MLP     | 0.591±0.006 | 18.886±0.070 | 48.980±0.268 | 19.895±0.145 | 0.732±0.009 | 0.200±0.002 |
> |   BiLSTM    | 0.628±0.001 | 19.078±0.024 | 57.030±0.583 | 22.673±0.145 | 1.008±0.018 | 0.209±0.002 |
> | Transformer | 0.646±0.001 | 19.379±0.076 | 60.969±0.452 | 22.195±0.709 | 1.079±0.023 | 0.198±0.001 |
> |  Fine-LDM   | 0.419±0.012 | 15.526±0.145 | 6.412±0.126  | 1.273±0.112  | 0.548±0.020 | 0.512±0.008 |
> | C2F-Decoder | 0.643±0.001 | 19.478±0.011 | 63.039±0.841 | 26.053±0.249 | 1.191±0.020 | 0.195±0.001 |
> |   C2F-LDM   | 0.454±0.021 | 15.883±0.151 | 6.102±0.365  | 1.504±0.323  | 0.520±0.021 | 0.530±0.010 |
>
> Brain2Speech
>
> |             |  **PCC↑**   |  **PSNR↑**   |   **FD↓**    |   **FAD↓**   |   **KL↓**   |  **CLAP↑**  |
> | :---------: | :---------: | :----------: | :----------: | :----------: | :---------: | :---------: |
> |     LiR     | 0.511±0.004 | 17.500±0.046 | 68.146±1.629 | 24.988±0.152 | 3.483±0.119 | 0.112±0.004 |
> |     MLP     | 0.409±0.008 | 16.389±0.122 | 75.174±0.708 | 27.983±0.890 | 4.153±0.021 | 0.094±0.004 |
> |   BiLSTM    | 0.526±0.000 | 17.688±0.050 | 92.172±0.465 | 33.442±0.257 | 4.187±0.015 | 0.074±0.002 |
> | Transformer | 0.526±0.000 | 17.690±0.055 | 74.048±1.083 | 27.526±0.280 | 3.817±0.030 | 0.041±0.002 |
> |  Fine-LDM   | 0.357±0.006 | 14.385±0.178 | 12.706±2.051 | 4.820±0.730  | 0.885±0.111 | 0.420±0.026 |
> | C2F-Decoder | 0.518±0.001 | 17.495±0.054 | 96.032±3.097 | 26.917±0.595 | 4.278±0.065 | 0.077±0.003 |
> |   C2F-LDM   | 0.393±0.012 | 15.260±0.179 | 9.726±1.528  | 4.623±0.508  | 0.616±0.062 | 0.471±0.017 |

---

> ### Author Response · Authors · 2024-11-22
> **Rebuttal by Authors [3/3]**
>
> >Why is C2F-LDM low for PCC and PSNR? [question 2.6]
>
> In the fields of signal generation and reconstruction, there exists a theoretical trade-off between perceptual quality and distortion metrics (such as PSNR) (Blau & Michaeli, 2018). The pursuit of better perceptual quality often leads to lower PSNR values, a phenomenon that has been widely validated in image generation research (Ledig et al., 2017; Wang et al., 2018). Direct decoding methods, which are optimized based on mean squared error, can indeed achieve higher PCC and PSNR, but the reconstruction results from these methods are often overly smooth and lack high-frequency details, resulting in poor perceptual quality.
>
> This trade-off is particularly evident in our reconstruction tasks. We choose to apply generative models with the goal of achieving reasonable signal reconstruction accuracy while maintaining high perceptual quality. Experimental results indicate that the approximately 0.4 PCC level achieved by our method is acceptable in the context of balancing perceptual quality and signal fidelity. Notably, our proposed C2F-LDM method outperforms the Fine-LDM baseline across all evaluation metrics, providing strong evidence for the effectiveness of the coarse-to-fine strategy.
>
> In lines 371-376 of the revised version, we have included a more detailed analysis of the metrics and further discuss the trade-off between perceptual quality and signal fidelity. We also consider introducing additional evaluation metrics in the future to comprehensively assess reconstruction quality.
>
> > Objective of Section 3.3 and 3.4 [question 2.7 weakness 3]
>
> Section 3.3 examines how coarse-to-fine decoding improves reconstruction quality, particularly focusing on the role of semantic enhancement. The study finds that when semantic decoding is suboptimal, the semantics of fine-grained features may be diminished. Building on the findings of Section 3.3, Section 3.4 investigates how to supplement semantic information through external prompts to address situations where semantic decoding is inadequate. Together, these sections support the core assertion of the paper: integrating multi-level decoding with semantics can enhance reconstruction outcomes.
>
> > Explanation of results in Figure 5(b) [question 2.7 weakness 3]
>
> The results indicate that the performance of the coarse-to-fine method varies across different subjects: while UTS01 and UTS02 show improved semantics, UTS03, UTS05, and UTS07 experience a decline. This variability is related to individual characteristics rather than indicating that "fine-grained decoding perform better for almost all experiments." Importantly, this phenomenon does not contradict the conclusions of the paper, as the core advantage of our method lies in the improvement of decoding accuracy (as shown in Figure 8) and the overall enhancement of reconstruction quality, rather than requiring superior performance on semantic decoding for all subjects.
>
> > Role of prompts [question 2.7 weakness 3]
>
> The prompt is designed as a supplementary mechanism rather than a requirement for performance enhancement in all situations. In the Brain2Music dataset, the effect of prompts declines, which aligns with the findings of Section 3.3: the Brain2Music dataset already exhibits satisfactory semantic decoding capability. In this case, additional semantic information may be redundant. Prompts are primarily intended for scenarios where semantic decoding is not optimal, which is consistent with our original design intention.
>
> > Voxel extraction [question 3.2]
>
> Brain2Sound: From a standardized template (HCP parcellation).
>
> Brain2Music and Brain2Speech: Using subject-specific parcellation based on anatomical and functional data (FreeSurfer).
>
> > Functional alignment [question 3.3]
>
> All datasets are functionally aligned through motion correction, slice timing correction, and cross-run alignment, etc.
>
> > Time window length [question 3.4]
>
> The current choice of a short time window is due to several factors, including the time resolution limitation of fMRI (TR), the need to maintain comparability with existing studies, and considerations of computational resources. While using longer windows may capture more temporal contextual information, it also significantly increases computational complexity and data requirements for training. In the future, we may explore how to better model the temporal dynamics of the BOLD signal and design model architectures that can handle variable-length time windows.

---

> > ### Author Response · Authors · 2024-11-22
> > **References**
> >
> > **Additional references:**
> >
> > - Naselaris, Thomas, et al. "Encoding and decoding in fMRI." *Neuroimage* 56.2 (2011): 400-410.
> > - Varoquaux, Gaël, et al. "Assessing and tuning brain decoders: cross-validation, caveats, and guidelines." *NeuroImage* 145 (2017): 166-179.
> > - Blau, Yochai, and Tomer Michaeli. "The perception-distortion tradeoff." *Proceedings of the IEEE conference on computer vision and pattern recognition*. 2018.
> > - Ledig, Christian, et al. "Photo-realistic single image super-resolution using a generative adversarial network." *Proceedings of the IEEE conference on computer vision and pattern recognition*. 2017.
> > - Wang, Xintao, et al. "Esrgan: Enhanced super-resolution generative adversarial networks." *Proceedings of the European conference on computer vision (ECCV) workshops*. 2018.

---

> > > ### Comment · Reviewer_DqUq · 2024-11-26
> > >
> > > Thank you to the authors for taking the time to respond to my numerous questions about the methods and the data and for making adjustments to your initial submission.
> > >
> > > In particular, the distinction between high-level/low-level and coarse/fine grain is now clearer, thanks to your response. However, explicitly making this distinction in the text would improve the general understanding of these terms throughout the manuscript. Thank you for further motivating the use of ridge regression, which is well-grounded; indeed, spatial organization seems out of scope for this paper but could be interesting for future work. I also appreciate the additional information about the results in Table 2 and the theoretical reasons for the model’s performance in terms of signal reconstruction.
> > >
> > > However, I remain unclear about the main objective of the paper and the authors’ claim that their model contributes to understanding the auditory system. While I understand that the model draws inspiration from the auditory system, in my opinion, the paper does not necessarily enhance our understanding of the auditory system (and does not seem to do so). Additionally, despite the authors’ responses to my questions about Sections 3.3 and 3.4, I still find these sections and results confusing and believe they do not support the main claim of the paper. Also, I would still question the use of the Transformer baseline; if I understood correctly, it is used on a (single) fMRI token.
> > >
> > > For these reasons, I maintain my original score.

---

### Author Response · Authors · 2024-11-25
**Response to Reviewers**

We sincerely appreciate all reviewers' detailed and constructive comments. We have carefully addressed each comment and made corresponding revisions in our updated draft. All modifications are shown in blue text for easy reference. In our point-by-point response below, we have indicated the specific line numbers where changes have been made to help track the revisions.

We believe these revisions have meaningfully improved our draft and look forward to any further feedback.

---

> ### Author Response · Authors · 2024-12-03
> **Final Response to Reviewers**
>
> We sincerely thank all reviewers for their feedback throughout this process. While we were unable to fully resolve all concerns in this revision, the reviewers' comments have provided clear direction for future improvements. We appreciate their time and consideration during this review process.

---

### Meta-Review · Area_Chair_PL3K · 2024-12-19

**Metareview:**

This submission provides a method for audio reconstruction from fMRI, presumably to be used for the study of those representations/encodings. The authors provide validation on three open datasets of reconstruction, and discussion of their work in the context of modelling in the auditory pathway.

There is a very healthy discussion of multiple aspects, but there are specific objections raised about motivation, contribution, and usefulness of the proposed work that remain unaddressed. In particular, reviewers `DqUq`, `JVh6`, and `Fjq3` all found that the modelling claims are not well supported, even after discussion. Quoting the response by `Fjq3` as summary:
> The motivation and utility of pre-trained model features in your method still also confuses me. The different components of your framework are inspired by the brain’s hierarchy, but you also claim that they ‘provide valuable insights into the human auditory system’ and reveal something of its ‘processing mechanisms.’ This reasoning seems a bit circular to me.

The other two reviewers raising this point responded similarly, highlighting a tension between engineering efforts and contribution to understanding (of the brain). This, in my opinion, speaks to a conceptual flaw in the work, or at the very least an unsuccessful attempt to provide two different things in one computational model, with an imperfect attempt to anneal their differences.

I find that there is merit in the present work, as evidenced by the relatively high scores from one reviewer, and based on comments from the lower scoring reviewers. However, I also find that there is still unrefined questions and conceptual issues that remain unanswered and unaddressed. Based on scores alone this work would be difficult to include; given it's particular flaws and the comments provided by reviewers, I think this work needs restructuring and possibly further meditation on its high level direction, choosing between reconstruction for reconstruction's sake (an engineering feat, though with dubious utility) or modelling interpretability, even noting the loaded nature of "interpretation".

**Additional Comments On Reviewer Discussion:**

I tend to agree with the discussion `DqUq` provides about Coarse or Fine features; these distinctions may not be entirely clear to a reader, depending on viewpoint. While the "coarse-grained" meaning "most abstract" appeases my own intuitions, it may be better for clarity's sake to rewrite this with more specific language about high-level, abstract, semantic features or low-level, local signal features.

I also agree with the concerns of `JVh6` about the additive Gaussian noise experiment:
>I would expect that the PSNR measured between the original audio and the audio + Gaussian noise would go down with increasing noise levels. However, it seems to increase in the table[.]

These are not rejection relevant concerns directly, but addressing them would improve the manuscript.

---

### Decision · Program_Chairs · 2025-01-22

Reject